# Reaction-based fluorogenic probes for detecting protein cysteine oxidation in living cells

Renan B. Ferreira[1], Ling Fu[2,3], Youngeun Jung[1], Jing Yang [2] & Kate S. Carroll [1]✉

'Turn-on' fluorescence probes for detecting $H_2O_2$ in cells are established, but equivalent tools to monitor the products of its reaction with protein cysteines have not been reported. Here we describe fluorogenic probes for detecting sulfenic acid, a redox modification inextricably linked to $H_2O_2$ signaling and oxidative stress. The reagents exhibit excellent cell permeability, rapid reactivity, and high selectivity with minimal cytotoxicity. We develop a high-throughput assay for measuring *S*-sulfenation in cells and use it to screen a curated kinase inhibitor library. We reveal a positive association between *S*-sulfenation and inhibition of TK, AGC, and CMGC kinase group members including GSK3, a promising target for neurological disorders. Proteomic mapping of GSK3 inhibitor-treated cells shows that *S*-sulfenation sites localize to the regulatory cysteines of antioxidant enzymes. Our studies highlight the ability of kinase inhibitors to modulate the cysteine sulfenome and should find broad application in the rapidly growing field of redox medicine.

Redox reactions and oxidative stress have been implied in the etiology of numerous diseases as well as in the aging process[1]. The modern era in translational redox medicine seeks to identify types, sources, metabolizers, and targets of oxidants in order to develop effective drugs and therapies for ROSopathies[2]. In this context, turn-on fluorescence probes also referred to as fluorogenic probes that measure oxidative stress in living cells have proven invaluable for redox-related biomedical research[3,4]. Analogous tools for detecting the reaction products between biological oxidants and proteins are grossly underdeveloped, a point exemplified by the lack of fluorogenic probes for detecting protein cysteine (cysteinyl) oxidation. The latter issue is especially striking since protein cysteines are the major target of oxidants originating from both endogenous and exogenous sources[5].

Sulfenic acid (Cys-SOH) is a central redox modification of protein cysteines and is inextricably linked to oxidant signaling and stress[6]. Sulfenic acid is generated by oxidation of a thiolate by reactive oxygen species (ROS) such as hydrogen peroxide ($H_2O_2$) produced during cellular signaling and metabolism or by hydrolysis of sulfenyl halides, and very polarized nitrosothiols and disulfides[7]. If stabilized by the protein microenvironment, the thiol-sulfenic acid pair can operate as a switch that is triggered by redox changes to regulate protein function, structure, and localization[8–11]. The electrophilic sulfur atom in sulfenic acid can also react with a protein or low-molecular-weight thiol to form a disulfide[12,13] or, under conditions of excess oxidative stress, can be oxidized further to sulfinic and sulfonic acids[14]. In either scenario, stabilized or as transient intermediate, sulfenic acids are key modifications in the domain of biological redox regulation.

Selective chemical detection of sulfenic acid is predicated on the chemical nature of this moiety in which the sulfur can behave as both a nucleophile and electrophile. The latter reactivity has been exploited by numerous groups to develop carbon-based nucleophilic probes for detecting sulfenic acid in proteins and cells[15]. The first fluorescent probe for sulfenic acid was reported in 2007[16], consisting of fluorescein attached at the C-4 position on 1,3-cyclohexanedione (Supplementary Fig. 1a). The nucleophilic C-2 carbon reacts selectively with sulfenic acid, but its application has been limited by the absence of "turn-on" fluorescence or fluorogenic response. In 2016, our group identified phenaline-1,3-dione as a chemical scaffold having

[1]Department of Chemistry, UF Scripps Biomedical Research, Jupiter, FL 33458, US. [2]State Key Laboratory of Proteomics, Beijing Proteome Research Center, National Center for Protein Sciences Beijing, Beijing Institute of Lifeomics, Beijing 102206, China. [3]Innovation Institute of Medical School, Medical College, Qingdao University, Qingdao 266071, China. ✉e-mail: kate.carroll@ufl.edu

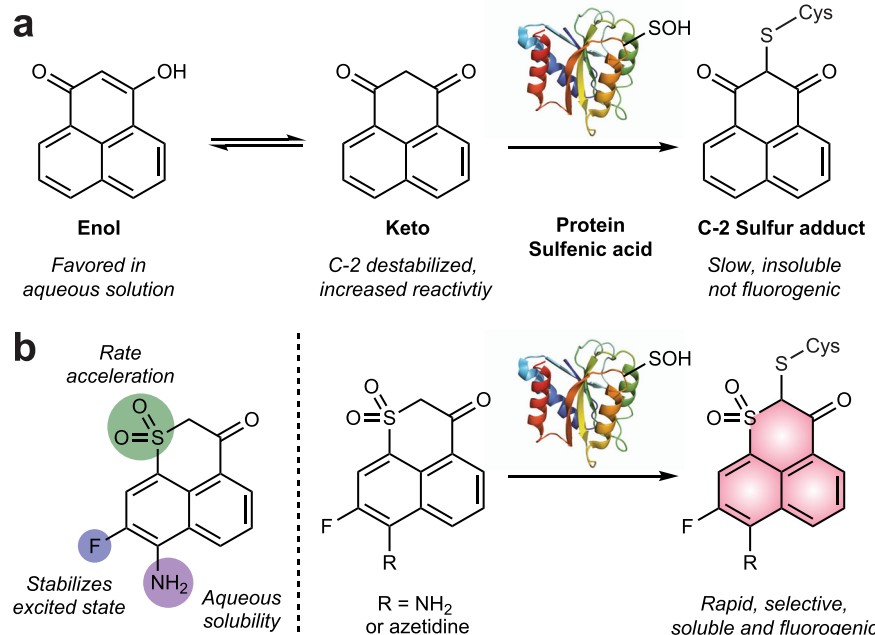

**Fig. 1 | Design strategy for developing fluorogenic probes for detecting sulfenic acid. a** General structure of phenaline-1,3-dione scaffold and the tautomerization between enol and keto forms. Reaction between the nucleophilic C-2 and electrophilic sulfenic acid sulfur shifts the keto-enol equilibrium toward the keto form, which has fluorogenic potential, but is slow and poorly soluble. **b** Colored circles denote the position for introducing electron-withdrawing groups to favor fluorogenic reaction with sulfenic acid (left). The resulting fluorogenic reaction-based probes are rapid, selective, and water soluble.

fluorogenic potential vis-à-vis its reaction with sulfenic acid[17] that was subsequently adapted in the design of a ratiometric fluorescent probe[18] (Fig. 1a and Supplementary Fig. 1b). However, exceedingly slow reaction rates of <0.01 M$^{-1}$s$^{-1}$ and the absence of turn-on fluorescence signal precluded any practical application.

To address the significant gap in chemical tools available that report on cysteinyl oxidation in real-time in living cells, here, we describe fluorogenic probes for detecting sulfenic acid (Fig. 1b and Supplementary Fig. 1c). The probes are reaction-based, exhibit excellent cell permeability, rapid reactivity, and high selectivity with minimal cytotoxicity. The chemical tools have enabled live-cell imaging studies to detect redox-dependent changes in cysteinyl oxidation. Adaptation to a 96-well plate format enabled high-throughput analysis of sulfenic acid in cells and this platform was used to screen a curated inhibitor library, targeting major families of the human kinome, as potential modulators of cysteinyl oxidation. These studies identified a cohort of GSK3 kinase inhibitors that elicited significantly increased sulfenic acid modifications that localized to the regulatory cysteines of proteins involved in response to oxidative stress. These studies highlight the utility of these chemical tools for facile fluorogenic detection of cysteinyl oxidation in a variety of formats with broad application in future studies of redox biology and drug pharmacology.

## Results

To create a fluorogenic probe for sulfenic acid, three key alterations of the phenaline-1,3-dione scaffold were envisioned: (i) replacement of a carbonyl group with the more electron-withdrawing sulfonyl group to increase the rate of reaction; (ii) appendage of an amino group to create an electron "push-pull system" system, and (iii) fluorination of the aromatic core to enhance absorption and fluorescence wavelengths (Fig. 1b). Optimized synthetic schemes for the resulting compounds are described within the Supplementary Methods in the Supplementary Information file. In brief, 6-bromo-2H-naphtho[1,8-bc] thiophen-2-one was prepared from naphthalene-1-thiol. The thiophenone core was hydrolyzed and dimethylated to produce both

thioether and ester functional groups and the thioether was oxidized to a sulfone group with mCPBA. The brominated sulfone intermediate was then used in amination reactions, followed by condensation in the presence of NaH to produce the C-nucleophile center. Finally, a fluorination step provided the desired compounds.

## Kinetic and fluorescence characterization of phenaline-1,3-dione derivatives

We first sought to assess the reactivity of phenaline-1,3-dione (**1**) and its analogs with an established small-molecule model for cysteine sulfenic acid, known as CSA[19,20] (Fig. 2a). Pseudo first-order rate constants were obtained for compound **1** and related analogs designed to evaluate features that modulate reactivity and fluorescence properties (Fig. 2b and Supplementary Fig. 2). Compound **1** exhibited modest kinetics 0.002 s$^{-1}$ and fluorination of this scaffold at the C-2 position (**2** and **3**) decreased reactivity by more than two orders of magnitude. Replacement of one carbonyl group with a sulfonyl moiety (**4**) increased the reactivity of the phenaline-1,3-dione core by more than 1000-fold. C-2 fluorination (**5**) of the parent sulfonyl compound enhanced reactivity while the introduction of a 7-amino group (**6**) decreased reaction rate. Although less reactive, the amino group enhanced the water solubility as compared to non-aminated analogs. Further modification by fluorination at C-6 (**7**) retarded kinetics by 10-fold. Finally, the 7-amino group was replaced by the cyclic amine, azetidine (**8**) affording a compound with an observed reaction rate of ~0.4 s$^{-1}$. Overall, the sulfonyl group dramatically increased nucleophilic reactivity towards the sulfenic acid electrophile, which was tempered somewhat by the installation of amino and fluorine groups, as expected.

In subsequent experiments, we evaluated the fluorescence properties of sulfonyl derivatives **4** through **8**. For this purpose, the reaction product between each analog and CSA was prepared and isolated from milligram-scale reactions. Fluorescence spectra of the resulting adducts were then recorded in organic solvent or aqueous solution and compared to that of non-adducted compounds alone (Fig. 2c, d, and Supplementary Fig. 3). CSA adducts of **4** and **5** exhibited weak

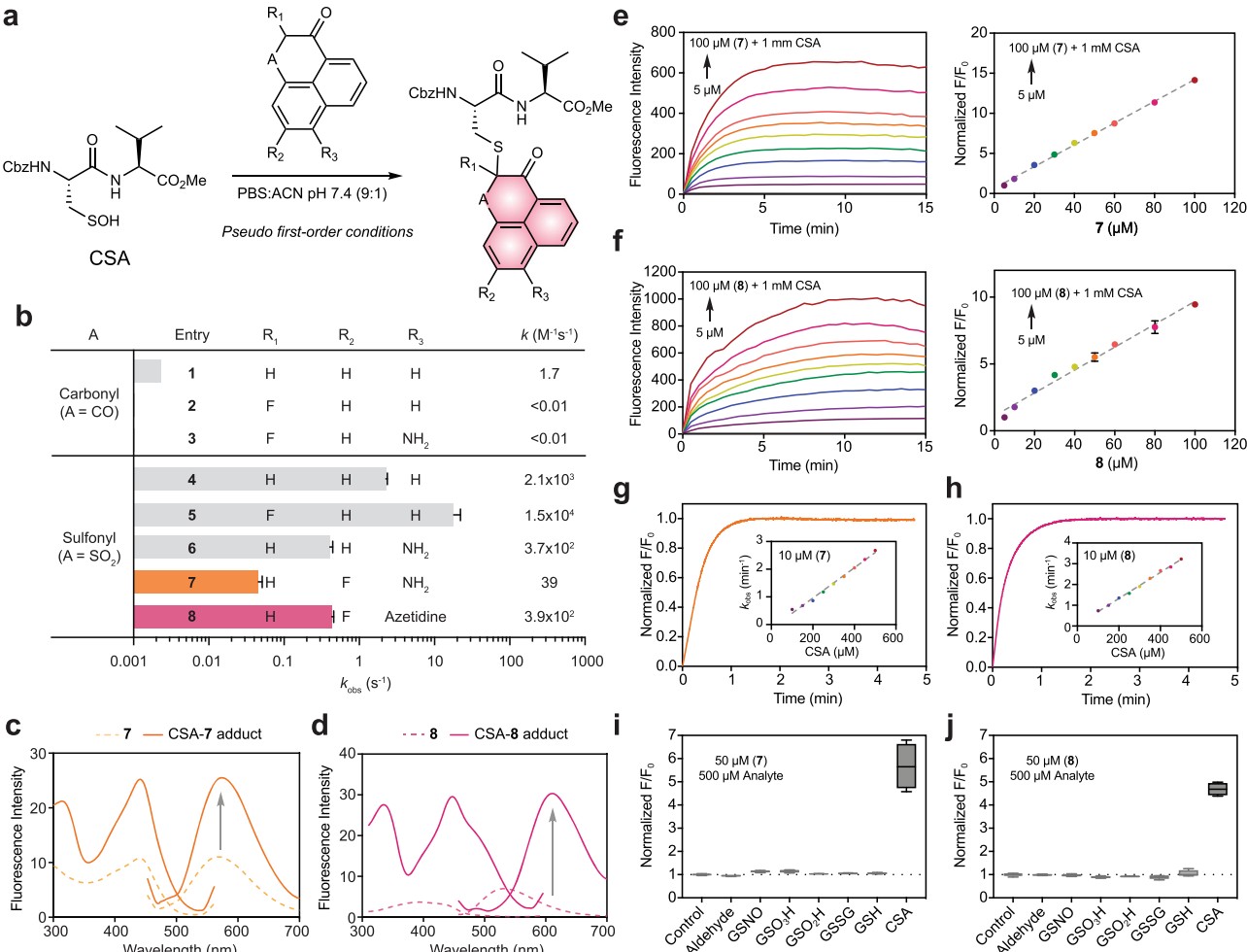

**Fig. 2 | Chemical modification of the phenaline-1,3-dione scaffold yields fluorogenic probes for detecting sulfenic acid. a** Rate constants were measured under pseudo-first-order conditions with small-molecule sulfenic acid model, CSA, in excess over phenaline-1,3-dione nucleophiles. **b** $k_{obs}$ ($s^{-1}$) for compounds **1** through **8** were determined by curve fitting to a single exponential function. Second-order rate constant, $k$ is extrapolated from $k_{obs}$ divided by the concentration of the limiting agent. Bars denote ±SEM. **c** Absorption and emission spectra of **7** and isolated CSA-**7** (250 μM) in PBS:ACN (9:1) pH = 7.4. **d** Absorption and emission spectra of **8** and isolated CSA-**8** (250 μM) in PBS:ACN (9:1) pH = 7.4. **e** Fluorescence intensity monitored over time for the reaction between **7** (5 to 100 μM) and CSA (1 mM) after excitation at 443 nm (left). Fluorescence intensity (15 min) is plotted versus the concentration of **7** (right). N = 3 independent experiments.

**f** Fluorescence intensity monitored over time for the reaction between **8** (5 to 100 μM) and CSA (1 mM) after excitation at 447 nm (left). Fluorescence intensity (15 min) is plotted versus the concentration of **8** (right). N = 3 independent experiments. **g, h** Rate constants were measured under pseudo-first-order conditions with CSA in excess over phenaline-1,3-dione nucleophiles. A representative trace from these reactions is shown. Linear fit of $k_{obs}$ ($min^{-1}$) plotted versus CSA concentration gives 91 ± 3 and 105 ± 2 $M^{-1}$ $s^{-1}$ for **7** and **8**, respectively (inset). N = 3 independent experiments. **i, j** Box and whisker plot of normalized fluorescence intensity ($F/F_0$) for reaction between **7** or **8** (50 μM) and various analytes (500 μM) after 1 h. Box plots show center line as median, whiskers show maxima and minima, and box limits show upper and lower quartiles. N = 4 independent experiments.

fluorescence signal and were essentially nonfluorogenic. Furthermore, compounds **4** and **5** were only sparingly soluble in water. Compound **6** showed improved solubility but gave only weak fluorescence signal before and after reaction with CSA. By contrast, CSA adducts of compounds **7** and **8** were fully soluble in PBS, exhibited bathochromic shifts of excitation and emission maxima as well as increases in fluorescence intensity compared to **7** or **8** alone, indicating that these compounds are fluorogenic for sulfenic acid (Fig. 2c, d). Specifically, the CSA-**7** adduct gave a slight bathochromic shift of excitation and emission maxima of 3 nm with no change in the Stokes shift. The bathochromic shift was more pronounced for the CSA-**8** adduct (53 and 75 nm for excitation and emission maxima, respectively) accompanied by an increase in the Stokes shift of 22 nm. The increase in fluorescence intensity was 2.3- for CSA-**7** and 11.4-fold for CSA-**8** (quantum yields increased by 2.1- and 2.6-fold, respectively) compared to **7** and **8** alone. UV-Vis spectroscopy analysis of **7** and **8** with their respective CSA adducts showed an increase in extinction coefficient of

1.7- and 6.4-fold, respectively (Supplementary Table 1 and Supplementary Figs. 4, 5). Finally, a bathochromic shift in the absorption maxima of CSA-**8** was also observed, in agreement with the shift observed in the fluorescence spectra. Comparison of data obtained for compounds **7** and **8** to compound **6** demonstrates that fluorination is essential for their fluorogenic response.

Next, we measured the fluorescence intensity of **7** and **8** in real-time reactions with CSA. Reactions containing **7** or **8** were excited at 443 and 447 nm, respectively. Fluorescence emission intensity increased over time and was linear with probe concentration (Fig. 2e, f). Pseudo first-order rate constants were then obtained at different probe concentrations to obtain second-order rate constants, which were 91 ± 3 and 105 ± 2 $M^{-1}$ $s^{-1}$ for **7** and **8**, respectively (Fig. 2g, h). To screen for potential side reactions, we evaluated the reactivity of **7** and **8** with a panel of potentially reactive biomolecules, such as aldehyde and disulfide electrophiles as well as related sulfur species including glutathione (GSH), glutathione nitrosothiol (GSNO), glutathione

sulfinic or sulfonic acid (GSO$_2$H or GSO3H). No significant reaction took place between **7** or **8** and any of the aforementioned sulfur compounds, including millimolar GSH (Fig. i, j and Supplementary Fig. 6a–c). Taken together, these studies indicate that compounds **7** and **8**, referred to hereafter as CysOx1 and CysOx2, exhibit reaction-based turn-on fluorescence, rapid reactivity, and high selectivity when evaluated in a small-molecule model of cysteine sulfenic acid.

### Evaluation of CysOx reactivity and fluorescence in complex biological settings

Encouraged by our success in the CSA model, we then moved on to more targets with greater biological relevance and complexity. Toward this end, we examined CysOx probe reactivity in C64,82 S glutathione peroxidase 3 (Gpx3; Fig. 3). This Gpx3 variant has one redox-sensitive cysteine 36 (C36) that is readily oxidized to sulfenic acid using stoichiometric amounts of hydrogen peroxide (H$_2$O$_2$) and has been well-validated as a model for the study of protein sulfenic acid reactivity[19,20]. Intact mass spectrometry (MS) analysis demonstrated that Gpx3 C36 sulfenic acid (Gpx3-SOH) formed the anticipated adduct with CysOx probes in high yield, while reduced Gpx3 (Gpx3-SH; Fig. 3b, c) or Gpx3 C36S were not modified (Supplementary Fig. 7a–c). Having verified the correct protein adducts by MS, we next evaluated the fluorescence spectra of isolated Gpx3-CysOx adducts compared to that of free CysOx probes. CysOx1 and CysOx2 exhibited a "turn-on" fluorescence response after respective excitation at 357 and 394 nm (Fig. 3d, e). Of the two probes, CysOx2 showed a larger fluorescence enhancement of up to fourfold. Fluorescence spectra of analogous real-time or in situ reactions gave similar results and also demonstrate that fluorogenic response is only observed when all three components, CysOx probe, H$_2$O$_2$, and Gpx3 are present in the reaction (Fig. 3f, g). Compared to CSA, the fluorescence spectra of Gpx3-CysOx adducts were blue-shifted by ~50 nm. Finally, the reaction products of CysOx probes and Gpx3 were visualized by in-gel fluorescence (Fig. 3h, I and Supplementary Fig. 7d, e). Intense signal was detected in CysOx-treated Gpx3-SOH but not Gpx3-SH or Gpx3 C36S, consistent with the findings in our intact MS analysis.

Having established that CysOx probes are fluorogenic and selective in small-molecule and protein models of sulfenic acid, we next assessed their ability to enter live cells and provide a fluorogenic readout of endogenous $S$-sulfenylated proteins. First, we examined whether CysOx probes could detect oxidation of epidermal growth factor receptor (EGFR) at C797, an established target of H$_2$O$_2$[8]. Cells expressing epitope-tagged wild-type or C797S EGFR were exposed to H$_2$O$_2$ in the presence of CysOx probes. Wild-type EGFR exhibited detectable fluorescence signal from CysOx1 or CysOx2, which was absent in C797S, as expected (Supplementary Fig. 8a, b). Next, we evaluated the ability of CysOx probes to visualize global protein $S$-sulfenation using glucose oxidase (GOX) at 0.2, 2, and 20 U/mL to provide continuous production of H$_2$O$_2$ in glucose-containing culture media[21]. In the absence of exogenous H$_2$O$_2$, CysOx probes were able to detect endogenous or basal H$_2$O$_2$ produced by enzyme oxidases and aerobic electron transport. Significant time- and GOX-dependent increases in fluorescence intensity, compared to the basal signal present in untreated cells, were also observed in the presence of CysOx1 or CysOx2 (Fig. 3a, b and Supplementary Fig. 9a–e). Fluorescence was distributed throughout the cytoplasm with no apparent co-localization and minimal cytotoxicity (Supplementary Figs. 10, 11). Live-cell labeling by non-fluorogenic analog **6** was also investigated. In contrast to CysOx probes, compound **6** failed to promote oxidant-dependent 'turn-on' fluorescence, instead giving weak signal throughout the duration of the experiment (Supplementary Fig. 12), consistent with our CSA studies. In-gel fluorescence analysis of lysates derived from cells incubated with CysOx probes showed concentration- and redox-dependent protein labeling by CysOx2 (Fig. 4c); signal from CysOx1 was weaker (Supplementary Fig. 9a–d), reflecting the inherent difference in their fluorogenic intensities. Chemoproteomic experiments also show that CysOx does not perturb the global cysteinome (Supplementary Data 1 and Supplementary Fig. 13). Collectively, these studies indicate that the reaction of CysOx probes and the electrophilic sulfur in sulfenic acid represents a viable strategy for fluorogenic detection of cysteinyl oxidation in the test tube and in real-time in living cells.

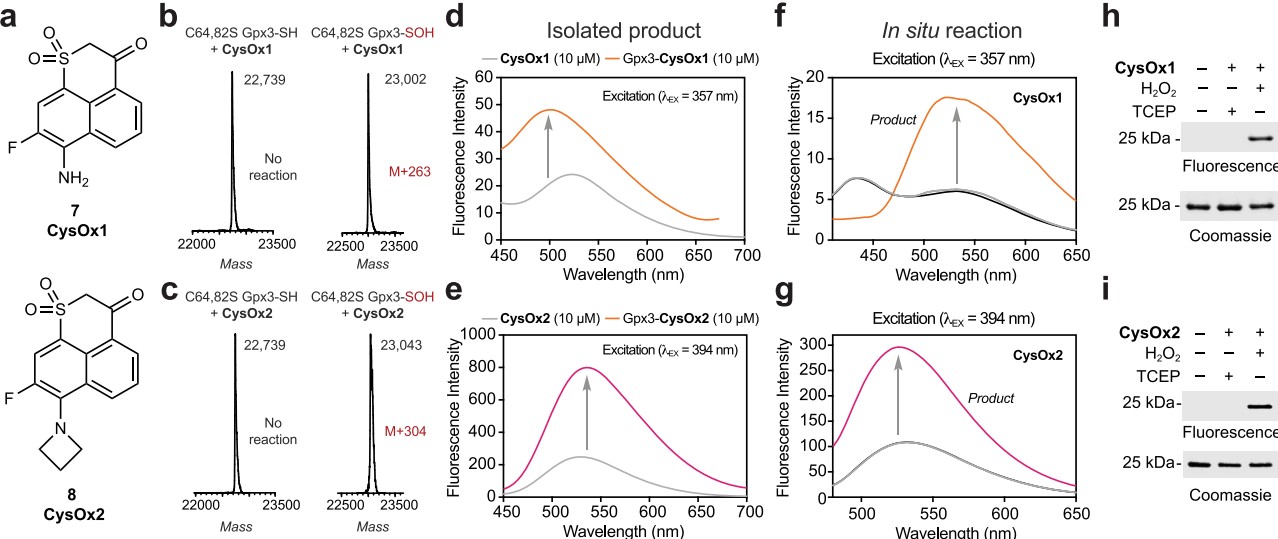

**Fig. 3 | CysOx1 and CysOx2 are reaction-based fluorogenic probes for sulfenic acid. a** Structures of CysOx1 and CysOx2. **b**, **c** Intact MS analysis of the reaction products of reduced or sulfenic acid forms of C64,82 S Gpx3 (10 μM) with CysOx1 or CysOx2 (1 mM) after 1 h in 50 mM HEPES pH 7.4. **d**, **e** Emission spectra of CysOx1 or CysOx2 alone (10 μM) compared to purified Gpx3-CysOx1 or Gpx3-CysOx2 (10 μM). Reactions with CysOx1 or CysOx2 were excited at 357 nm and 394 nm, respectively. Spectra were recorded in 50 mM HEPES pH 7.4. **f**, **g** Emission spectra of CysOx1 or CysOx2 alone (5 μM) or in combination with H$_2$O$_2$ (15 μM) or Gpx3 (5 μM) or H$_2$O$_2$ (15 μM) and Gpx3 (5 μM). Only the reaction containing all three components gives a fluorogenic product. Reactions with CysOx1 or CysOx2 were excited at 357 and 394 nm, respectively. Spectra were recorded in 50 mM HEPES pH 7.4. Analysis by intact MS indicates that ~75% Gpx3-SOH is labeled by CysOx probes under these conditions. **h**, **i**, In-gel fluorescence analysis of reaction products between Gpx3 (10 μM) and CysOx1 or CysOx2 (1 mM) with or without H$_2$O$_2$ (15 μM) or TCEP (1 mM) after 1 h in 50 mM HEPES pH 7.4. $N = 2$ independent experiments for each probe.

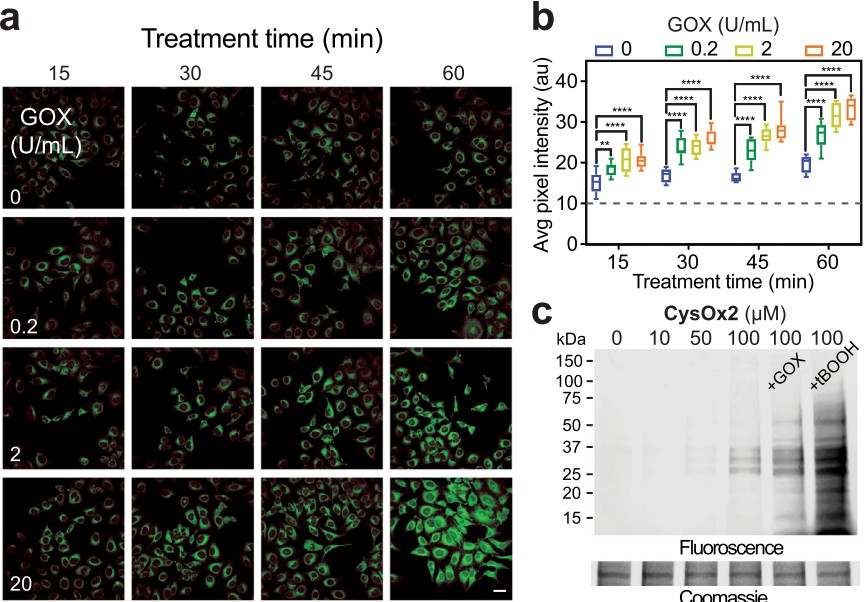

**Fig. 4 | Cell permeable CysOx probes for no-wash live-cell microscopy of sulfenic acid in cells. a** No-wash live-cell confocal images of HeLa cells at different time points after the addition of CysOx2 (10 μM) and GOX at the indicated concentrations (0–20 U/mL). $\lambda_{ex}$ = 458 nm; scale bar: 50 μM. Representative images are shown from $N$ = 3 independent experiments. **b** Box and whisker plot of the average pixel intensities from panel **a**. Data are representative of ten independent readings from five different frames. Bars denote ±SEM. Variance was analyzed by two-way ANOVA test. At 15 min: **$P$ = 0.0099 and ****$P$ < 0.0001 when compared against cells treated with probe only. At 30, 45, and 60 min: ****$P$ < 0.0001 when compared against cells treated with probe only. **c** Representative in-gel fluorescence analysis of lysates derived from HeLa cells incubated with CysOx2 and GOX (20 U/mL) or $t$-BOOH (500 μM). $N$ = 2 independent experiments. Fluorescence lane intensity is quantified and shown in Supplementary Fig. 9e.

## Application of CysOx2 to identify kinase inhibitors that modulate cysteinyl oxidation in cells

Crosstalk between cysteine oxidation and other post-translational modifications such as phosphorylation had been reported[22,23] but the interplay between kinase inhibition and cysteinyl oxidation remains unclear. To further showcase the utility of fluorogenic probes in the detection of cysteinyl oxidation and address the question above, we subsequently adapted CysOx2 for use in a cell-based 96-well plate assay. After probe concentration and treatment times were optimized (Supplementary Fig. 14) we screened a curated library of kinase inhibitors was in HeLa cells. Analysis of the resulting data revealed a threefold or greater increase in fluorescence for 8% (12 out of 154 compounds) of kinase inhibitors as compared to vehicle-treated cells (Fig. 5a and Supplementary Data 2). Protein $S$-sulfenation was modulated by inhibition of RTKs as well as select kinases from the AGC and CMGC families, including the multitasking Ser/Thr kinase GSK3 (Fig. 5b). Upon closer inspection, three of the ten compounds associated with the greatest increase in fluorescence intensity were identified as GSK3 inhibitors (SB-216763, BIM-IX, and Bio; Fig. 5a–c and Supplementary Fig. 15), suggesting a relationship between this idiosyncratic kinase and protein $S$-sulfenation. In support of this hypothesis, the ability of each GSK3 inhibitor to induce probe reactivity in cells correlated with loss of β-catenin phosphorylation (Fig. 5g, h and Supplementary Fig. 16a, b) and activation of glycogen synthase (GS; Supplementary Fig. 16c). Furthermore, application of the most GSK3-selective inhibitor identified in this screen, SB-216763 promoted a strong increase in fluorescence, equivalent to exogenous peroxide (Fig. 5c). The ability of SB-216763 to increase protein $S$-sulfenation was also confirmed through fluorescence imaging microscopy (Fig. 5f, g) and in-gel fluorescence analysis (Supplementary Fig. 17).

## Site-specific proteomic ID of cysteines that undergo $S$-sulfenation in GSK3 inhibitor-treated cells

Intrigued by the ostensible connection between GSK3 inhibitor treatment and cysteinyl oxidation, we quantified site-specific changes in the $S$-sulfenome using BTD-based chemoproteomics[24,25] (Fig. 6a). Lysates obtained from HeLa cells treated with or without GSK3 inhibitors were labeled by BTD, a clickable chemical probe for detecting sulfenic acid. Probe-labeled proteomes were then cleaved into peptides by trypsin, reacted with light or heavy azide-tagged photocleavable biotin via click chemistry, and combined equally. Next, probe-modified peptides were enriched on streptavidin resin, eluted by UV light exposure, and analyzed by LC-MS/MS. The heavy to light ratio ($R_{H/L}$) calculated for each BTD-labeled site reports the relative amount of $S$-sulfenation in inhibitor-treated samples versus controls (Fig. 6b); changes in protein expression should also be kept in mind. On average, we quantified ~300 sulfenic acid sites for each GSK3 inhibitor (Supplementary Data 3). SB-216763 produced the most significant perturbation to the $S$-sulfenome with 16% of sites exhibiting a significant increase (Fig. 6c; $R_{H/L} \geq 1.5$, $p < 0.01$), followed by BIM-IX (11%) and then Bio (2%) inhibitors;' these data are in excellent agreement with the findings in our fluorescence kinase screen. SB-216763 increased sulfenic acid modification of several well-known examples of redox-regulated cysteines (Fig. 6d) and gene ontology (GO) analysis indicates that the sub-sulfenome perturbed by GSK3 inhibition was significantly enriched ($p < 0.01$) in redox-related processes (Fig. 6e).

## Discussion

The fluorogenic character of CysOx probes stems, at least in part, from the fluorescence quenching effect of the phenaline-1,3-dione keto tautomer, in which nearly parallel carbonyl groups give the largest large dipole moment[26]. The reaction of the phenaline-1,3-dione scaffold at nucleophilic C-2 with the electrophilic sulfur atom in sulfenic acid serves to stabilize the fluorogenic enol tautomer. A second key contributor to the fluorescence of these probes is the electron-withdrawing fluorine atom attached to the naphthalene core, which serves to stabilize the excited state[27]. A final feature of CysOx probes is the push-pull system wherein the electron-donating amino group is in conjugation with fluorine via the intervening π-system[28].

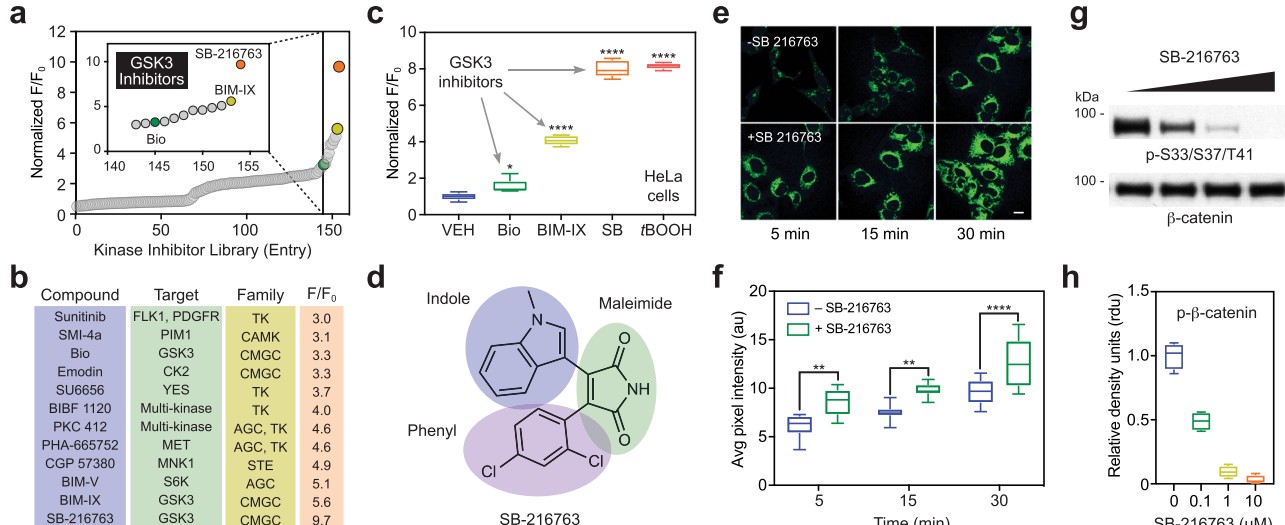

**Fig. 5 | Multi-well-based screening identifies kinase inhibitors that elevate cysteinyl oxidation in cells. a** Relative fluorescence intensity of HeLa cells treated with various kinase inhibitors (10 μM) and CysOx2 (50 μM) after 1 h at 37 °C. **b** Twelve compounds demonstrated a relative increase in fluorescence intensity of 3-fold or greater. The primary target and kinase family for each compound is listed. **c** Three GSK3 inhibitors were identified among the top ten hits: Bio, BIM-IV (BIM) and SB-216763 (SB). These inhibitors were purchased, manufacturers' purity was verified by LC-MS, and then rescreened in HeLa cells at 1 μM in the presence or absence of CysOx2 (50 μM). The concentration of t-BOOH control was 2 mM. Box plots show center line as median, whiskers show maxima and minima, and box limits show upper and lower quartiles. The variance was analyzed by one-way ANOVA test. *$P$ = 0.035 and ****$P$ < 0.0001 when compared against cells treated with vehicle only. **d** General structure of disubstituted maleimide-based GSK3 inhibitors. **e** No-wash live-cell confocal images of HeLa cells at different time points after the addition of CysOx2 (5 μM) in the presence or absence of SB-216763 (0.1 μM). λ_ex = 458 nm; scale bar: 50 μM. Representative images are shown from $N$ = 3 independent experiments. **f** Average pixel intensities from panel **e**. Data are representative of ten independent readings from five different frames. Error bars represent ±SEM. The variance was analyzed by two-way ANOVA test. At 5 min: **$P$ = 0.0010; at 15 min: **$P$ = 0.0025; at 30 min: ****$P$ < 0.0001 when compared to cells treated with probe only. Box plots show center line as median, whiskers show maxima and minima, and box limits show upper and lower quartiles. **g** Cells were treated with vehicle or with 0.1, 1 or 10 μM (left to right) SB-216763 for 30 min and then 50 nM calyculin. **h** Phospho-β-catenin-S33/S37/T41 and total β-catenin band densities were determined for each blot and expressed as relative density units (rdu). Bars denote ±SEM. $N$ = 3 independent experiments. Box plots show center line as median, whiskers show maxima and minima, and box limits show upper and lower quartiles.

Another aspect of this study was to apply CysOx technology to identify kinase inhibitors that modulate *S*-sulfenation in cells. The curated screening library used in this study includes inhibitors of lipid, receptor, and non-receptor tyrosine, serine/threonine, and dual-specificity kinases[29]. Several interesting patterns emerged from this screen. For instance, among inhibitors associated with a significant increase in cellular *S*-sulfenation, about 40% targeted kinases in the RTK family. This is consistent with reports from our lab and others indicating that growth factor signaling is intimately linked to ROS metabolism in cells[5,8]. The balance of "hits" within this cohort target members of the AGC and CMGC families and, of special interest, three of these inhibit GSK3 function[30]. GSK3 is an unusual kinase in that it is constitutively active and its substrates, over 100 are known, need to be pre-phosphorylated by another kinase, and GSK3 is inhibited as opposed to activated by the two pathways known to converge on GSK3, the insulin and Wnt pathways[31]. Our observation that cellular *S*-sulfenation increases with GSK3 inhibition are corroborated by earlier reports that such compounds, including SB-216763, increase cellular ROS[32,33]. The elucidation of such patterns highlights the ability of CysOx probes to identify small-molecules that modulate protein *S*-sulfenation and is relevant to many fields including covalent targeting of semi-conserved cysteines which is a growing strategy in drug design[34,35].

To independently verify the connection between GSK3 inhibition and cysteinyl oxidation in our kinase screen we performed chemoproteomic analysis using the selective and clickable probe for sulfenic acid, BTD. Cell treatment with each of the three inhibitors increased cellular *S*-sulfenation significantly with SB-216763 exhibiting the largest increase, followed by BIM-IX and then Bio. The simplest explanation to account for the difference in *S*-sulfenation is that while all three compounds are GSK3 inhibitors, only SB-216763[36] has been reported to be selective for this kinase[37] (see also Supplementary Fig. 15). BIM-IX[38] and Bio[39] inhibit protein kinase C (PKC) and cyclin-dependent kinases (cdks), respectively, and are thus, partitioned across additional multiple cellular targets. The selectivity profile is fully consistent with the potency of these compounds in functional GSK3 assays, including loss of β-catenin phosphorylation and GS activation. More generally, comparing the chemoproteomics findings herein to our earlier work[14] reveals a different *S*-sulfenation pattern between kinase inhibitor- and $H_2O_2$-treated cells. For example, in HeLa cells exposed to $H_2O_2$, most *S*-sulfenation sites remain unchanged ($R_{H/L} ≤ 1.5$). In fact, cysteines in redox-regulated proteins, like peroxiredoxin 6 (PRDX6) C47, exhibit a decrease in this modification with a concomitant increase in *S*-sulfinylation ($-SO_2H$)[14]. SB216763, on the other hand, upregulates sulfenic acid modification at more than 40 unique sites including PRDX6, indicating that GSK3 inhibition produces more targeted changes in cysteine oxidation than might be expected from non-specific diffusion of $H_2O_2$ throughout the cell.

Regarding GSK3 and redox regulation, inhibition of this kinase is well-known to activate the transcription factor, Nrf2 which regulates an array of detoxifying and antioxidant defense gene expression[40]. Beyond this direct connection between cellular redox and GSK3, a separate study shows that the Nrf2 pathway can also be triggered by inactivation of thioredoxin and glutathione-glutaredoxin systems[41], which are critical reducing mechanisms in eukaryotes. Also consistent with this report, administration of SB-216763 is associated with protection against DNA damage[42]. Our finding that GSK3 inhibitors, like SB-216763, increase *S*-sulfenation of redox-active regulatory cysteines in thioredoxin (Trx), thioredoxin reductase (TR), glutathione reductase (GR) and glutaredoxin (Grx) may represent a KEAP1-independent means to activate Nrf2; however, future research will be required to dissect this ostensible relationship in greater molecular detail.

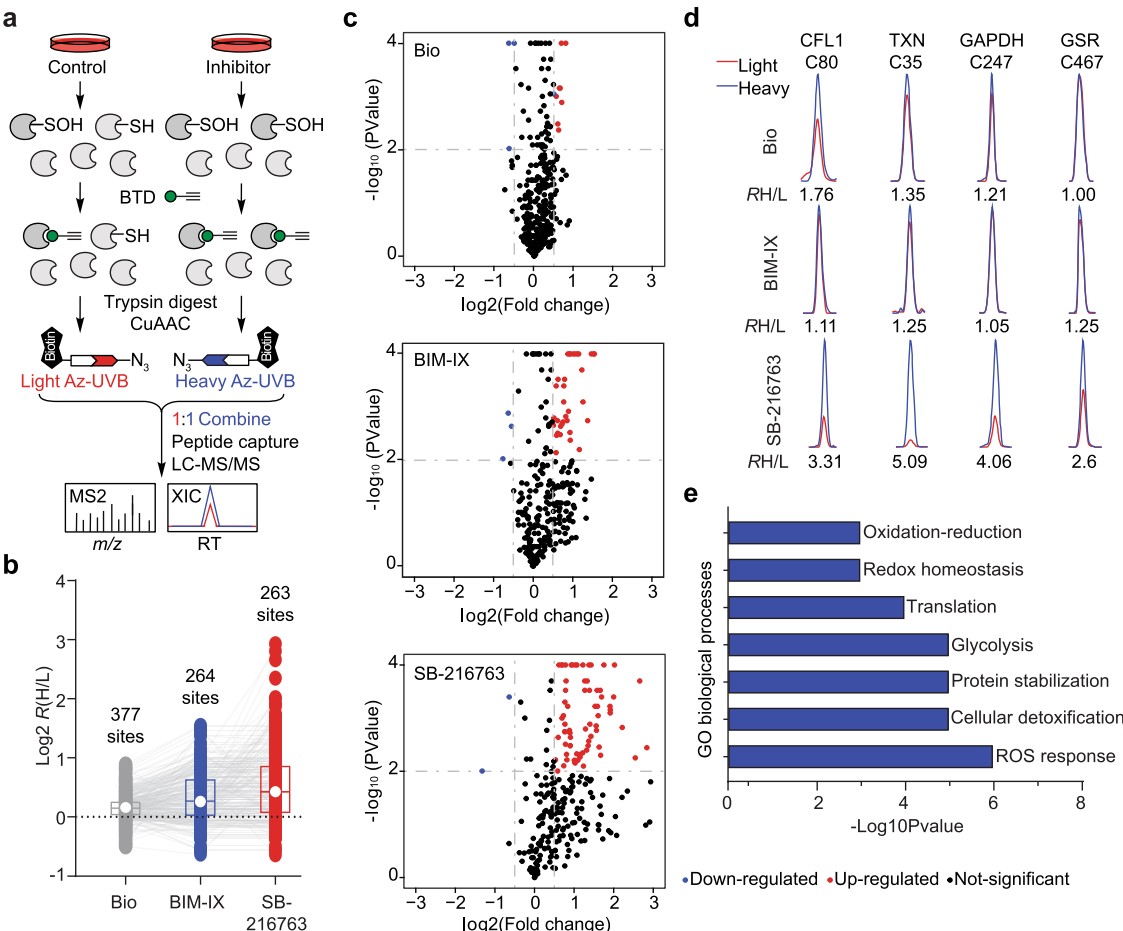

**Fig. 6 | GSK3 inhibition selectively perturbs the *S*-sulfenome. a** Workflow for quantitative *S*-sulfenome analysis. HeLa cells were treated with inhibitor (1 μM, 1 h) or vehicle as indicated. Lysates were prepared and treated with BTD (5 mM, 1 h) to label sulfenic acids. BTD-labeled proteomes with or without inhibitor treatment were digested and conjugated to heavy or light azide-tagged photocleavable biotin, respectively. Light and heavy labeled samples then were combined, clicked to biotin, enriched on streptavidin resin, and analyzed by LC-MS/MS. A heavy to light ratio ($R_{H/L}$) calculated for each BTD-labeled cysteine residue reports its relative level in inhibitor-treated samples versus controls. **b** Box plot showing a side-by-side comparison of results from GSK3 inhibitors Bio, BIM-IX, and SB216763. from two biological replicates. **c** Volcano plots showing the log2 values of the ratio between GSK3 inhibitor (heavy) and control (light) channels and the -log10(P) of the statistical significance in a two-sample *t*-test for all quantified cysteines. Sulfenation events that were significantly changed with GSK3 inhibition are shown in red ($R_{H/L} > 1.5$, $P < 0.01$). **d** Representative extracted ion chromatograms (XICs) showing changes in BTD-labeled peptides from the proteins as indicated. The profiles for light- and heavy-labeled peptide are shown in red and blue, respectively. The average $R_{H/L}$ values calculated from two biological replicates are displayed below each individual XIC. **e** GO enrichment of the sub-sulfenome perturbed by GSK3 inhibition.

Fundamentally, our study shows that kinase inhibition by small-molecules alters cysteine *S*-sulfenation in cells. This observation connects to a growing list of studies that highlights the link between kinase activation, target phosphorylation and protein cysteine modifications. New proteomic technologies such as CysPAT elegantly link EGFR signaling, phosphorylation to reversible cysteine modifications, including the catalytic cysteine site of protein tyrosine phosphatases (PTPs)[43]. A chemoproteomic method relating serine/threonine phosphorylation levels to changes in cysteine reactivity has recently been reported[44]. Another proteomic method, termed Cys-BOOST, shows changes in nitrosothiol (SNO) levels in response to early nitrosative stress during neuro(axono)genesis signaling[45]. Cross-talk between S-nitrosylation and phosphorylation is also documented[46]. GSK3 itself is a target of cysteine modification and *S*-nitrosylated GSK3β has been reported to translocate to the nucleus to phosphorylate nuclear substrates[47].

As with all technologies, it is worth noting current limitations. At present, the CysOx fluorophore presents low brightness and quantum yields when compared to other dyes, such as fluorescein and BODIPY. This is reflected in the low quantum yields (0.9–2.4%) observed for the reaction product of sulfenic acid and CysOx probes. While this issue

did not hinder application of CysOx probes in the present work, attempts to further miniaturize the cell-based assay to a 384-well plate format were unsuccessful. Future iterations of these tools will involve improving probe brightness and dynamic range. We also note that the ability of CysOx probes to monitor subtle endogenous signaling processes was not examined in this study but will be an important area of future research.

## Methods

### General spectroscopy

Fluorescence spectra for chemical and biochemical studies were recorded on an Agilent Cary Eclipse Fluorescence Spectrophotometer and analyzed using Eclipse software v1.2. Absorbance spectra were recorded on an Agilent Cary 300 Bio UV-Visible Spectrophotometer and analzyed using WinUV software v4.20.

### Kinetic studies

Stock solutions of compounds **1**–**8** were prepared in DMSO at a final concentration of 100 mM and then diluted in PBS. A stock solution of dipeptide sulfenic acid (CSA)[19] was prepared in acetonitrile at a final

concentration of 100 mM. To a solution of compound (2 mL), was added a solution of CSA (1 mL) and incubated at room temperature. Final concentrations were as follows: Compounds **1–8**: 25, 100, or 250 μM; CSA: 2.5, 10, or 25 μM adhering to pseudo first-order conditions. Aliquots (300 μL) of the reactions were collected at regular intervals and quenched by the addition of formic acid (100 μL). The resulting samples were analyzed LC-MS using an Agilent 1220 Infinity LC system coupled to a 6120 Quadrupole running OpenLAB CDS software vA.03.02.024. The amount of product plotted versus time to obtain $k_{obs}$ using KaleidaGraph software v 4.1.1. Reactions of **7** or **8** were also measured at 10 μM final concentration and CSA at 100-500 μM final concentration in PBS:ACN (9:1) pH = 7.4 were prepared and the fluorescence signal was immediately recorded for 5-10 min. Fluorescence versus time was plotted to obtain $k_{obs}$ which were plotted against the concentration of CSA to obtain second-order rates constants.

### Fluorescence quantum yield (φ) calculation

Lucifer Yellow (LY, $\varphi_{LY} = 0.21$ in water) was used as a general reference for all analytes due to proximity of its excitation $\lambda_{max}$ (428 nm) to those from unreacted CysOx1 or CysOx2 and their covalent CSA adducts (443–447 nm). Three dilute solutions of each compound with absorbances between 0.01 and 0.1 at the $\lambda_{max}$ were prepared in PBS. Emission spectra of these solutions using each analyte's $\lambda_{max}$ as excitation wavelength were recorded and areas under the emission spectra were integrated. Emission spectra of the LY reference were obtained using different excitation wavelengths corresponding to each analyte's $\lambda_{max}$ to ensure the same excitation photon density between reference and analyte. Linear regression of area of fluorescence versus absorbance provided slopes ($m$), which were applied in the following equation to calculate φ for each analyte, where the subscript LY refers to data obtained with Lucifer Yellow:

$$\phi = \phi_{LY} \times \frac{m}{m_{LY}}$$

### Intact mass spectrometry (MS) of C64,82S Gpx3

C64,82S Gpx3 was thawed at 0 °C and reduced with 50 mM (2 M, 5 μL) DTT for 20 min on ice. C64,82 S Gpx3-SH was exchanged into labeling buffer (50 mM HEPES, 100 mM NaCl, pH = 7.4) using P-30 spin columns (Bio-Rad). The C64,82 S Gpx3-SH solution was diluted in labeling buffer to give a final protein concentration of 10 μM in a reaction volume of 200 μL. 8 μL of a 25 mM DMSO solution of CysOx1 or CysOx2 was added, followed by 3.0 μL of 1 mM $H_2O_2$ or 700 mM TCEP in ddH$_2$O. Reaction mixtures were incubated at room temperature for 1 h, spin filtered using P-30 spin columns, and resolved on a C4 column (Grace Vydac, Cat. No. 214TP3405) connected to an Agilent 1220 Infinity LC. The molecular mass of CysOx-labeled C64,82 S Gpx3 was measured on a Thermo Scientific LTQ XL linear Ion trap mass spectrometer. Spectra were acquired in the positive ion mode and deconvoluted using MagTran software v1.0.

### In situ reaction of C64,82S Gpx3 with CysOx1 or CysOx2

Solutions of C64,82S Gpx3-SH (20 μM) and CysOx1 or CysOx2 (20 μM) were prepared in labeling buffer and stored on ice. The Cary Eclipse software was set to Kinetics mode using the fluorescence excitation and emission maxima observed for Gpx3-CysOx adducts (357/498 nm for Gpx3-CysOx1 and 394/535 nm for Gpx3-CysOx2). C64,82 S Gpx3-SH (50 μL) and CysOx probe (50 μL) were mixed in a cuvette. Next, $H_2O_2$ (100 μL of 30 μM in labeling buffer) was added and fluorescence measurements were acquired in triplicate.

### In-gel fluorescence analysis of CysOx-labeled proteins

Gpx3 labeled with CysOx1 or CysOx2 was diluted in reducing SDS-PAGE load dye, boiled for 5 min, and stored on ice. For HeLa cell lysates

derived from CysOx-treated cells 20 μg of protein was diluted to a final volume of 30 μL in non-reducing load dye, boiled for 5 min, and stored on ice. SDS-PAGE precast 4–15% or 4–20% gradient gels (Bio-Rad) were electrophoresed in Tris-Glycine-SDS running buffer at 80 V for 10 min and then at 120 V for 60 min. Gels were then rinsed with ddH$_2$O and imaged using the ChemiDoc MP imaging system (Bio-Rad). Settings were as follows: Fluorescein (epi-blue 460–490 nm excitation and 577–613 nm emission filter). After fluorescence imaging, gels were stained in Commassie Brilliant Blue solution for 5 min at room temperature, destained, and then visualized using ChemiDoc MP. Uncropped and unprocessed scans for these and all other blots are provided in the Source Data file.

### Fluorescence cell imaging of live cells treated with CysOx probes

HeLa cells were purchased from ATCC® (#CCL-2). HeLa cells (seeding population: $8.0 \times 10^4$ cells/well) were incubated in EMEM (+10% FBS) in 6-well glass bottom plates at 37 °C. At 70–80% confluence, media was aspirated, and cells were washed twice with PBS and serum-free EMEM was added to wells. After incubation for 16 h at 37 °C the media was aspirated, and cells were washed twice with PBS. For GOX treatment, cells were treated with CysOx probe and GOX at the indicated concentrations in PBS containing 0.1% DMSO and glucose (1 mg/mL). At the indicated time points, wells were analyzed using an Olympus FluoView IX81 confocal microscope using FV10-ASW software v3.0. For *t*BOOH treatment, oxidant was added at the indicated concentrations in EMEM for 10 min at 37 °C followed by the addition of CysOx probe at the indicated concentration for additional 15 min at 37 °C. The media was then aspirated, cells washed twice with PBS, fresh PBS was added, and wells were analyzed as above. Fluorogenic CysOx signal was visualized using the 458 nm laser channel, filtered using a SDM560 dichroic mirror, followed by BA505-605 band pass filter. Five frames were recorded per condition. Images were analyzed using ImageJ software v1.52a, where the pixel intensity of the cellular cytoplasmic regions was measured with at least 10 measurements per condition.

### Cell viability

HeLa cells were incubated in EMEM supplemented with 10% FBS on black 96-well plates with clear bottoms ($2 \times 10^4$ cells/well) at 37 °C. Once the cells acquired a confluence of 80–90% (after 24-48 h), the media was aspirated, cells were washed twice with PBS and fresh EMEM (lacking FBS) was added. The cells were incubated at 37 °C overnight. The cells were then washed twice with PBS, and PBS (90 μL) was added to each well. The wells were treated with a serial 1:4 dilution of CysOx1 or CyxOx2 (5000, 1250, 313, 78.1, 19.5, 4.88, 1.22, 0.305, 0.076, and 0.019 μM final concentrations) in PBS containing 1% DMSO (total volume/well = 100 μL, 10 wells per condition). One set of ten wells was treated with PBS containing 1% DMSO lacking probe as a viability control for untreated cells. The cells were incubated at 37 °C for 1 h and 100 μL of CellTiter-Glo® reagent (Promega, Cell-titer-Glo® Luminescent Cell Viability Assay, Cat. No. G7570) was added to each well. Plates were mixed for 2 min on an orbital shaker and then incubated at room temperature for 10 min. Luminescence was recorded in a Molecular Devices SpectraMax M5 plate reader using SoftMax Pro software v5.4 with an integration time of 1 s per well.

### Colocalization analysis of CysOx probe labeling with ER and Mitochondria

HeLa cells ($8.0 \times 10^4$ cells/well) were incubated in EMEM (+10% FBS, supplemented with penicillin/streptomycin) in six-well plates with a glass bottom for imaging (Chemglass, Cat. No. CLS-1812-006). Culture media was aspirated, cells were washed twice with PBS, EMEM lacking FBS was added to the wells, and cells were incubated at 37 °C for 16 h. Culture media was aspirated, and cells were washed twice with PBS. Wells were treated with CysOx1 or CysOx 2 (50 μM final concentration) and an organelle-selective dye (Invitrogen, ER-tracker red, Cat. No.

E34250 or Cell Signaling Technology, Mitotracker Deep Red FM, Cat. No. 8778 S) in EMEM with 0.1% DMSO for 1 h at 37 °C. Selected wells were also treated with $H_2O_2$ (300 μM final concentration). After incubation, culture media was aspirated, cells were washed twice with PBS, fresh PBS was added, and analyzed using an Olympus FluoView IX81 confocal microscope using FV10-ASW software v3.0.

### Preparation of lysate from HeLa cells incubated with CysOx probes

HeLa cells ($2 \times 10^8$ cells/well) were scraped into lysis buffer (50 mM HEPES, 100 mM NaCl pH 7.4 containing 0.1% SDS, 1% NP-40, 1X EDTA-free protease inhibitor, and 200 U/mL of catalase) at 4 °C for 20 min. Cell lysates were clarified by centrifugation at 14,000 rpm for 20 min at 4 °C and passed through Zeba spin desalting columns (Pierce, 7 K MWCO) to remove excess probe and protein concentration was determined by BCA assay.

### Detection of S-sulfenation in EGFR

HeLa cells ($2 \times 10^8$ cells/well) were transfected with wild-type or C797S EGFR pCMV6-XL-4[48]. After 24 h, cells were treated with $H_2O_2$ (500 μM) and then CysOx1 or CysOx2 (50 μM) for 1 h at 37 °C. Lysates were prepared as above. EGFR was immunoprecipitated from 500 μg cell lysate (1 mg/ml) with 2 μg mouse anti-EGFR conjugated agarose (Santa Cruz Biotechnology, Cat. No. sc373746 AC, clone A-10) overnight at 4 °C. Resin was collected by centrifugation and washed with cold RIPA buffer (1×) and cold PBS buffer (2×). Bound proteins were eluted by boiling with reducing load dye, resolved by SDS-PAGE, and analyzed for CysOx labeling by in-gel fluorescence as detailed above. For Western blot detection of total EGFR, samples were resolved by SDS-PAGE, transferred to PVDF membrane, and blocked with 3% BSA. Membranes were washed with TBST, and immunoblotting was performed with using anti-EGFR antibodies conjugated to HRP (Santa Cruz Biotechnology, Cat. No. sc-373746 HRP, 1:100). Blots were developed with chemiluminescence (Pierce) and imaged by film. Data were quantified by densitometry using ImageJ v1.52a.

### Multi-well cell-based screening assay to identify kinase inhibitors that modulate S-sulfenation

The 'Kinase Screening Library' was purchased from Cayman Chemical (Cat. No. #10505). The library consists of two plates and contains ~160 curated selective and non-selective kinase inhibitors in a 96-well plate format as 10 mM stock solutions in DMSO. This curated library includes inhibitors of a wide range of lipid, receptor and non-receptor tyrosine, serine/threonine, and dual specificity kinases including those belonging to the ROCK, activin-like kinase (ALK), GSK3, PKC, PDGFR, VEGFR, Src, MAPK, CDK, and PI3K families, among many others. It offers coverage of more than 70 distinct kinases and kinase families, as well as numerous additional kinase isoforms and individual kinases within target families. Detailed information on inhibitor selectivity can be found on the manufacturer's website. HeLa cells were incubated in EMEM supplemented with 10% FBS in black 96-well plates with clear bottoms (seeding population: $2 \times 10^4$ cells/well) at 37 °C. At 90% confluence (48 h), media was aspirated, cells washed twice with PBS and serum-free EMEM was added to the wells. After incubation for 16 h at 37 °C media was aspirated, cells were washed twice with PBS, and then PBS (90 μL) was added to each well. CysOx2 (50 μM) and kinase inhibitor (10 μM) in PBS containing 1% DMSO was added each well for a total volume 100 μL per well. Cells were then incubated for 1 h at 37 °C. Select control wells did not receive kinase inhibitors and were instead treated with either PBS (negative control) or tBOOH (200 μM, positive control). Following the incubation period, PBS (100 μL) and extracellular fluorescence quencher (20 μL, Solution C of Beta-Lactamase Loading Solutions Kit from Life Technologies, Cat. No. K1048) were added to each well prior to measurement in a Molecular Devices SpectraMax M5 plate reader and analyzed using SoftMax Pro software

v5.4. The fluorescence parameters of CysOx2 were selected according to the maxima measured in the fluorescence spectra (excitation/emission: 447/606 nm). Preliminary screening to identify hits was performed in duplicate and compounds exhibiting more than a 2.5-fold increase in fluorescence were subsequently confirmed in quintuplicate according to the same procedure.

### Measurement of β-catenin phosphorylation and GS activity

HeLa cells ($2 \times 10^8$ cells/well) were treated with vehicle or with 0.1, 1 or 10 μM SB-216763 for 30 min and then with 50 nM calyculin at 37 °C. Lysates were prepared as described above, separated by SDS-PAGE, and transferred to PVDF membrane. Immunoblotting was performed using primary and secondary antibodies at the indicated dilutions in TBST: β-catenin (Cell Signaling, Cat. No. 8480, 1:1000) or phospho-β-catenin-S33/S37/T41 (Cell Signaling, Cat. No. 9561, 1:1000) and goat anti-rabbit IgG-HRP (Calbiochem, Cat. No. 12-348, 1:25,000). Blots were developed with chemiluminescence, imaged by film, and quantified as above. To measure GS activity, serum-starved HeLa cells were treated for 60 min with 10 μM Bio, BIM-IX or SB-216763 or DMSO vehicle. Cells were harvested, non-denaturing lysates prepared, and supernatants assayed for GS activity using the GCS activity assay kit (Cat. No. AK0138-100T-96S, Sunlong) by monitoring the oxidation of NADH oxidation to $NAD^+$ at 340 nm.

### Live cell HeLa labeling with clickable sulfenic acid probe BTD

HeLa cells ($2 \times 10^4$ cells/well) were incubated for 1 h with different concentrations of SB-216763 (0, 0.01, 0.1, 1, 5, and 10 μM final concentrations) and 1 mM BTD in PBS supplemented with 1% DMSO for selected times. Culture media was aspirated, and cells were washed three times with cold PBS. The supernatant was aspirated, and cells were incubated in 300 μL of lysis buffer (50 mM HEPES, 100 mM NaCl pH 7.4 containing 0.1% SDS, 1% NP-40, 1X EDTA-free protease inhibitor, and 200 U/mL of catalase) at 4 °C for 20 min. Cell lysates were clarified by centrifugation at 16,000 rpm for 20 min at 4 °C. Protein concentration was determined by BCA assay. Lysates were diluted to a concentration of 1 mg/mL (100 μg in 100 μL). Click reactions were performed in each sample: 2 μL of TAMRA-azide (5 mM) was added, followed by addition of a pre-mixed solution of 2 μL $CuSO_4$ (12.5 mM in $ddH_2O$) and 5 μL of BTTP (10 mM in $ddH_2O$). Then, 2 μL sodium ascorbate (125 mM in $ddH_2O$) was added into each sample, which was incubated at room temperature with gentle shaking for 1 h. Reactions were quenched by addition of EDTA (2 μL, 50 mM in $ddH_2O$). Samples were resolved by SDS-PAGE and then analyzed by in-gel fluorescence in the 520 nm channel to observe TAMRA fluorescence in an Azure Sapphire Biomolecular Imager. The gel was then incubated with GelCode™ Blue Safe Protein Stain (Thermo Scientific) for 30 min and then with water for additional 30 min and imaged.

### Chemoproteomic S-sulfenome analysis

HeLa cells ($2 \times 10^8$ cells/well) treated with or without GSK3 inhibitor were harvested, lysed in pre-chilled NETN buffer [50 mM HEPES (pH 7.6), 150 mM NaCl, and 1% IGEPAL] supplemented with protease and phosphatase inhibitors (Thermo Scientific, Cat. No. A32961) containing 200 U/mL catalase (Sigma-Aldrich), and then incubated with 5 mM of BTD at 37 °C for 2 h with rotation and light protection. The resulting probe-labeled samples were incubated with 40 mM iodoacetamide at 37 °C for 1 h with light protection. To remove excess small molecules, proteins were precipitated by methanol-chloroform (aqueous phase/methanol/chloroform, 4:4:1 (v/v/v)). The precipitated proteins were resuspended with 50 mM ammonium bicarbonate containing 0.2 M urea and digested with sequencing grade trypsin (Promega) at a 1:50 (enzyme/substrate) ratio overnight at 37 °C. The tryptic digests were desalted, evaporated to dryness, resuspended in a water solution containing 30% acetonitrile (ACN) and subjected to CuAAC ligation[24]. After 2 h incubation at room

temperature, the light and heavy reaction mixtures were then combined, purified by strong cation exchange (SCX), and then subjected to the enrichment with streptavidin beads for 2 h at room temperature. Streptavidin beads were washed with 50 mM NaAc (pH4.5), 50 mM NaAc containing 2 M NaCl (pH4.5), and deionized water twice each with vortexing and/or rotation to remove non-specific binding substances, then resuspended in 25 mM ammonium bicarbonate, photo-released under 365 nm UV light for 2 h at room temperature with magnetic stirring. The supernatant was collected, dried under vacuum, and stored at −20 °C until further analysis. LC-MS/MS analyses were performed on a Q Exactive plus instrument (Thermo Fisher Scientific). Peptide samples were reconstituted in 0.1% formic acid and pressure-loaded onto a 2-cm microcapillary precolumn packed with C18 (3-μm, 120 Å, SunChrom, USA) operated with an Easy-nLC1000 system (Thermo Fisher Scientific). The precolumn was connected to a 12-cm 150-μm-inner diameter microcapillary analytical column packed with C18 (1.9-μm, 120 Å, Dr. Maisch GebH, Germany) and equipped with a homemade electrospray emitter tip. The spray voltage was set to 2.0 kV and the heated capillary temperature to 320 °C. The LC gradient consisted of 0 min, 7% B; 14 min, 10% B; 51 min, 20% B; 68 min, 30% B; 69–75 min, 95% B (A = water, 0.1% formic acid; B = MeCN, 0.1% formic acid) at a flow rate of 600 nL/min. MS1 spectra were recorded with a resolution of 70,000, an AGC target of 3e6, a max injection time of 20 ms, and a mass range from $m/z$ 300 to 1400. HCD MS/MS spectra were acquired with a resolution of 17,500, an AGC target of 1e6, a max injection time of 60 ms, a 1.6 $m/z$ isolation window and normalized collision energy of 30. Peptide $m/z$ that triggered MS/MS scans were dynamically excluded from further MS/MS scans for 18 s. Raw data files were searched against Homo sapiens Uniprot canonical database using pFind studio software v3.1.2[49]. Precursor ion mass and fragmentation tolerance were set as 10 ppm and 20 ppm, respectively. The maximum number of modifications and missed cleavages allowed per peptide were both set as three. For all analyses, mass shifts of +5.9949 Da (methionine oxidation) and +57.0214 Da (iodoacetamide alkylation) were searched as variable modifications. For site-specific mapping of probe-modified sulfenic acid sites, mass shift of +418.131 ($C_{19}H_{22}N_4O_5S$) for BTD was searched as variable modification. A differential modification of 6.0201 Da on probe-derived modification was used for stable-isotopic quantification. The FDRs were estimated by the program from the number and quality of spectral matches to the decoy database. The FDRs at spectrum, peptide, and protein level were <1%. Note that identifications based on single PSM (peptide-spectrum match) was allowed, which were manually inspected[50]. For those identified peptides containing multiple cysteine residues, at least three modification-specific fragment ions were required for unambiguous site localization. Quantitative analyses from two biological replicates were performed using pQuant software v1.0[51].

### Statistics and reproducibility
Measurements were taken from distinct samples. Data are reported as mean ± SEM unless otherwise indicated in the figure legend. Statistical analyses were performed using one- or two-way ANOVA with Tukey's honest significant difference test correction for multiple comparisons using GraphPad Prism v9.4. For all analyses, $p < 0.05$ was considered significant.

### Reporting summary
Further information on research design is available in the Nature Research Reporting Summary linked to this article.

### Data availability
The data supporting the findings from this study are available within the manuscript and its supplementary information. The chemoproteomic data sets have been deposited to the ProteomeXchange

Consortium via the PRIDE partner repository with the dataset identifiers PXD029176. Source data are provided with this paper.

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

## Acknowledgements

We thank Dr. Louis Scampavia and Dr. Timothy Spicer (UF Scripps Biomedical Research) for discussions on high-throughput screening. We thank Dr. Laura Bohn and Dr. Edward Stahl (UF Scripps Biomedical Research) for training and technical support in fluorescence microscopy. Finally, we thank Longqin Sun and Tuo Zhang (Beijing Qinglian Biotech Co., Ltd) for their technical support. This work was supported by grants from the Natural Science Foundation of China (21922702) and the State Key Laboratory of Proteomics (SKLP-K201703 and SKLP-K201804) to J.Y., and NIH (GM102187 and CA222849) to K.S.C.

## Author contributions

R.B.F., K.S.C., and J.Y. designed the experiments, analyzed data. K.S.C. wrote the manuscript with input from all authors. R.B.F. synthesized and characterized all compounds, performed kinetic studies, fluorescence measurements, confocal imaging, cellular, and kinase inhibition studies. Y.J. performed biochemical studies with Gpx3C36S. J.Y. and L.F. performed chemoproteomic experiments and analyzed the data.

## Competing interests

The authors declare no competing interests.
