## [Peer Review File · Nature Communications]

REVIEWER COMMENTS

Reviewer #1 (Remarks to the Author):

The authors developed fluorescent probes that detect sulfenic acid derived from cysteine oxidation in proteins. Sophisticated fluorescent probes have been highly desired to visualize this posttranslational modification. In this study, a phenaline-1,3-dione chemical scaffold was redesigned to develop “fluorogenic” probes that enhance fluorescence in response to protein sulfenic acid. The introduction of sulfonyl and fluoro groups contributed to improve binding kinetics, while amination increased water solubility. One of the probes, CSA-8 or CysOx2, showed significant fluorescence enhancement upon binding to CSA. Contrary to the expectation from the CSA results, protein sulfenic acid enhanced fluorescence modestly (up to only 4-fold). Fluorescence imaging demonstrated fluorescence detection of cysteine oxidation was possible in living cells using CysOx2. The probe was applied to investigate the effect of kinase inhibitors on the cysteine oxidation. While live-cell imaging results were shown, the background signals are surprisingly intense. Even in the absence of the ROS generating enzyme or reagents, fluorescence signals were observed. In addition, the signals were gradually enhanced in that condition. Although the fluorescence results of protein sulfenic acid was presented, it is questionable whether it is really useful for investigating redox phenomena, which is not given strong oxidizing environment unlike the ideal condition used in this study, since the background signals are too high to neglect. Another concern is whether GSK3 inhibition is really linked with protein sulfenylation, since no proof for it is provided. First of all, no data that shows inhibition of phosphorylation by GSK3 was presented. Moreover, it is not also clear whether the inhibition of GSK3 really perturbed protein sulfenylation, since there is a possibility that the inhibitor may have nonspecific interaction with other proteins or biomolecules. Therefore, I do not think that the conclusion is fully supported by the data shown in this study. Due to the low performance of the probe and the lack of the data supporting the conclusion, this study is more suitable for publication in more specialized journal. The other minor comments were described in the below.

1. Supporting figure S4 and 5: In addition to the compound names CysOX1 and 2, those of compounds 7 and 8 should be described together. CysOX1 and 2 are not presented at this step in the text.
2. Fluorescence gel are marked as “fluorescein” or “fluorocein”. These should be corrected to “Fluorescence”.
3. Figure numberings of Fig 5 and 6 in the text are wrong.
4. Why is the background signal of CysOX2 so high in the fluorescence spectrum of protein adduct, while the CSA adduct spectrum shows only low background signals?

Reviewer #2 (Remarks to the Author):

Ferreira et al. describe the development and application of a novel fluorogenic probe to label and detect cysteine sulfenic acid (CSA). Sulfenic acid is one of several important Cysteine posttranslational modifications which play central roles in redox signaling in health and disease, but which are still not well-studied because of the lack of appropriate tools. The authors have previously reported a probe for CSA which, however, suffered from slow reaction rates and the absence of turn-on fluorescence – shortcoming that have been addressed in the present manuscript. They use their novel probe in a 96-well format to study the impact of kinase inhibitors on CSA and identified a set of kinase inhibitors that induce an increase in CSA. Using a complementary chemoproteomics approach, they could confirm these findings and identify target proteins and Cys sites.

The manuscript is well-written and clearly structured. A few comments should be addressed.

1) To test the specificity of the probes, the authors used small potentially reactive biomolecules (fig 2 i,j) showing almost no reactivity except for CSA. GSH is still low but slightly elevated – given the concentration of GSH in cells, could this be a potential issue, considering that PTMs typically have a low stoichiometry? Also, the text reads glutathione sulfinic acid had been used, while the figure reads methanesulfinic acid. What was the control?

2) Encouraged by the high specificity, the authors then used their compounds to analyze CSA in a more complex system, namely Gpx3. Here, they demonstrate that reduced Gpx3 does not react with their probes, but what if this C36 was a sulfinic acid version (way more complex system than in fig 2), will the probe still be as selective as shown for the small biomolecules?

3) Figures 3b and 3c: What is the yield (“high yield” is very unspecific), is the CSA-Gpx3 quantitatively labeled? The authors should mention the efficiency of the labeling directly in the figure. On the same note, in the kinase experiment the probe concentration was considerably lower --- how is the labeling efficiency in this case? This should be discussed.

4) The probes show minimal cytotoxicity, which is great. I wonder what side effects the treatment with such probes will have though. What would a differential proteomic study with and without probe treatment (as performed in the kinase inhibitor screen) reveal, and what would this implicate?

5) Please give some more details about the kinase inhibitor library, e.g., what is known about the specificity of these inhibitors?

6) Huang et al. (PMID 27281782) have previously described the impact of kinase stimulation on Cys modifications, including the modulation of Cys in the active sites of phosphatases. Same holds true for Cys-nitrosylation (PMID 31097712). This should be added to the discussion, as these results are in good agreement with the finding that kinase inhibition affects Cys redox modifications.

7) Page 10, line 182: “suggesting a relationship between this idiosyncratic kinase and protein S-sulfenylation.” Page 12, line 203 “These findings suggest that redox changes induced by GSK3 inhibition might be more specific as opposed to simple diffusion of H₂O₂.”

It is known that the inhibition of GSK, e.g., by SB-216763 is involved in the regulation of oxidative stress. This should be considered and mentioned here. Especially the second statement does not surprise in this context.

8) Figure 5: What exactly does it mean that “the purity of the inhibitors was verified by LC-MS”? That the purity was as claimed by the manufacturer?

9) The chemo-proteomics results part would benefit from 2-3 more lines on what has actually been done. Also, the data should be uploaded to a repository, as is standard in the field. Considering a 1-hour treatment, did the authors consider the impact of protein expression level changes induced by the treatment? For phosphorylation it has been shown by the Gygi lab that a substantial share of the supposedly regulated phosphorylation events were indeed rather changes on the level of protein expression. Although I do not consider this as a major issue here, it would be good to briefly discuss this possibility.

10) Figure 6: When looking at the suppl. data, it appears that some sites have only been quantified in a single replicate --- it is unclear whether these are included in the figure and in the results in general. If so, they should be removed and the discussion should focus on findings that have been replicated. In general, the suppl. table is not easy to read and should be changed to a more user-friendly format (e.g., individual replicate ratios should be in individual cells, not separated by “|”). Did the authors use any site localization assessment to ensure that sites in peptides with multiple Cys residues are correctly assigned?

11) The manuscript title should be changed to “cysteine sulfenic acid” rather than “cysteine oxidation” as this is actually misleading, unless the authors clearly explain how this same probe(s) can be used for other Cys oxidative PTMs?

12) Suppl. methods: Why was sodium ascorbate used in the BTD assay, please explain. Shouldn't this reduce Cys nitrosylation?

Reviewer #3 (Remarks to the Author):

This manuscript by Ferreira et al. describes a new class of “turn on” fluorogenic probes to detect sulfenic acid modification of protein cysteine residues in living cells. The probes were developed from previously reported non-fluorogenic sulfenic acid-reactive phenaline-1,3-diones. The authors provide good evidence that the new reagents provide a substantial fluorescence increase and change in spectral properties upon reaction specifically with sulfenic acid but not other cysteine oxidation products. Following on proof-of-concept experiments showing that the compounds fluoresce in cells treated with

oxidizing agents, they perform a focused screen of small molecule kinase inhibitors and identify a number that appear to promote protein sulfenic acid modification in cells. Of particular interest were several GSK3-beta inhibitors, and the authors go on to show through sulfenic acid-targeted proteomics that GSK3-beta inhibitors indeed broadly induce this modification in proteins, albeit with some specificity for specific Cys residues. Overall this is a well-executed and insightful study that reports tools that will be broadly useful for the study of oxidative protein modification.

I do have a few suggestions for improving the manuscript:

1. The consistent identification of GSK3-beta inhibitors from the small molecule screen is intriguing. However, as no kinase inhibitor is exquisitely selective, one worries about potential off-target effects. Can the authors provide some reassurance that the effect of these inhibitors to promote sulfenic acid modification is on-target? For example, the three compounds vary in their efficacy in promoting Cys modification. Does this effect correlate with their potencies as GSK3-beta inhibitors? Can the authors verify target engagement by correlating a marker for GSK3-beta inhibition (e.g. loss of beta-catenin phosphorylation) with efficacy in inducing probe reactivity in their cell culture system?

2. In the proteomics experiments, cutoffs used to score modification induced by inhibitor treatment are too loose – for the less potent inhibitors a “hit” is anything with a modification ratio >1.0, which of course is going to include a lot of sites that are not significantly upregulated. It would be preferable if they used both an effect size cutoff AND a significance cutoff ($-\log p > 1.3$ is standard). Doing so will almost certainly reduce the number of consistent hits, which should make the GO analysis more robust. It would also be of interest to know how the GO analysis compares with prior studies that have used general oxidizing agents to induce sulfenic acid modification – in other words, are GSK3-beta inhibitors having a selective effect as suggested by the authors, or does they simply induce a low level of oxidative stress (which would still be interesting).

Minor point:

On p. 10, the references to Fig 5 are mislabeled as Fig 4.

Reviewer #4 (Remarks to the Author):

(General comments to authors)

Authors developed fluorogenic probes for detecting sulfenic acid, called CysOx1 and CysOx2 which can visualize cysteinyl oxidation in real-time in living cells. Using CysOx2 in a cell-based 96 well plate assay with kinase inhibitors, they demonstrated that TK, AGC, and CMGC family kinases including GSK3 modulate S-sulfenylation. Chemoproteomic mapping of sulfenic acid-modified cysteines in GSK3 inhibitor-treated cells show that site of Cys oxidation localize to regulatory Cysteines within key components of antioxidant defense systems. Study is novel and well performed for biochemical characterization of newly developed CysOx1 and CysOx2 probes. However, there are several major weaknesses. First, evaluating CysOx reactivity to Gpx3-S-sulfenylation was not sufficient. Second, functional significance of GSK3-sensitive S-sulfenylation of identified antioxidant enzymes in H₂O₂-treated cells as well as relationship between phosphorylation and S-sulfenylation of antioxidant enzymes are not clear. Third, many data are not quantified with lack of number of experiments. Fourth, there are lots of miss labeling of figures. Therefore, authors should address the following concerns by performing additional experiments to increase the quality of this paper.

(Specific comments to authors)

1. Figures 3de and 3hi: It is important to show that H₂O₂-induced Gpx3-SOH formation is inhibited in Gpx3-C36A mutant, but not Gpx3-WT, detected by CysOx1 and CysOx2 probes using fluorescence spectra and in gel fluorescence analysis.
2. Figure 4C: Data should be quantified by showing averaged graphs with dot-plots and statistical analysis.
3. Before investigating effects of kinase inhibitors on cysteinyl oxidation in cells, authors should confirm their previous observation that H₂O₂ induces oxidation of EGFR at Cys797 (Cell Chem Biol 2016. 23:837-) in cells transfected with EGFR-WT vs. EGFR-Cys797A using CysOx1 and CysOx2 probes.
4. In Line 196-199, it is mentioned that “the sub-sulfenylome perturbed by GSK3 inhibition was enriched in many biological processes involved in antioxidant responses including regulatory cysteines of thioredoxin, thioredoxin reductase, glutaredoxin and glutaredoxin reductase (Fig. 5de). These data are presented in Figure 6. It is important to provide the number of experiments and statistical analysis.
5. Authors should provide the functional significance of s-sulfenylation of identified antioxidant enzymes in GSK-dependent biological responses such as cell survival or cell proliferation or cell migration in H₂O₂-treated HeLa cells by transfecting mutant proteins in which reactive Cys residue is mutated to Ala or Ser.
6. Since GSK is kinase, direct substrates should be phosphorylated. Thus, it is important to clarify the relationship between phosphorylation and s-sulfenylation of antioxidant enzymes, too.
7. Supplementary Figure 7: It is important to provide the lists of s-sulfenylated proteins localized in ER and mitochondria, respectively.
8. Supplementary Figure 12: It is important to provide the number of experiments and statistical analysis.

(Minor comments to authors)

1. There are lots of mislabeling of Figures, which should be corrected.

Reviewer comments and revisions:

Reviewer 1:

- 1. “While live-cell imaging results were shown, the background signals are surprisingly intense. Even in the absence of the ROS generating enzyme or reagents, fluorescence signals were observed.”***

The observation of fluorescence signal in the absence of exogenous ROS generating enzymes or oxidants is not unexpected. In fact, cells continuously produce endogenous H₂O₂. NOXs and other related enzymes produce H₂O₂ for signaling. In the mitochondria, aerobic respiration also produces significantly amounts of H₂O₂ at steady-state. Consistent with these normal processes, rigorously validated chemoproteomics has been used to identify thousands of cysteinyl sulfenic acids in unstimulated cells also referred to as “basal” S-sulfenylation¹. Since CysOx probes are reaction-based, endogenous cysteinyl sulfenic acids – those that occur naturally in untreated cells – will be present and represent basal cysteinyl sulfenic acids. The addition of exogenous ROS generating enzymes or H₂O₂ to cultured cells results in a significant, detectable increase in cysteinyl sulfenylation as detected by CysOx fluorogenic probes (Figs. 3-4). To educate non-experts in the field, we include new text on p. 9-10 of the revised manuscript.

- 2. “Another concern is whether GSK3 inhibition is really linked with protein sulfenylation since no proof for it is provided.”***

Respectfully, we disagree with the assertion above. Live-cell labeling, lysate analysis and chemoproteomics with validated CysOx probes demonstrate that administration of validated GSK3 inhibitors increases cellular S-sulfenylation. At the same time, although the ability of Bio, BIM-IX and SB-216763 to inhibit GSK3 is well-documented, our original manuscript did not directly demonstrate that these compounds effect GSK3 in our experimental system. To strengthen the link between Bio, BIM-IX and SB-216763, GSK3 activity and S-sulfenylation, we performed additional experiments and analyses as follows:

- 1) GSK3 substrate phosphorylation:** As suggested by Reviewer 3, GSK3-inhibitor target engagement is substantiated by dose-dependent loss of β -catenin phosphorylation. GSK3 inhibitor potency in these studies is in excellent agreement with the observed change in S-sulfenylation measured by quantitative proteomics (Bio<BIM-IX<SB-216763). These new data presented as Fig. 5g,h and Supplementary Fig. 16a,b in the revised manuscript.
- 2) Glycogen synthesis:** GSK3 inhibition by the above-mentioned compounds is further corroborated by activation of glycogen synthase (GS) activity. These new data are presented as Supplementary Fig. 16c in the revised manuscript.
- 3. “Supporting figure S4 and 5: In addition to the compound names CysOX1 and 2, those of compounds 7 and 8 should be described together. CysOX1 and 2 are not presented at this step In the text.”**

This has been corrected.

- 4. “Fluorescence gel are marked as “fluorescein” or “fluorocein”. These should be corrected to “Fluorescence”.”**

This has been corrected.

- 5. “Figure numberings of Fig 5 and 6 in the text are wrong.”**

This has been corrected.

6. “Why is the background signal of CysOX2 so high in the fluorescence spectrum of protein adduct, while the CSA adduct spectrum shows only low background signals?”

CSA is a tripeptide. Gpx3 is a complex macromolecule. The background signal in the Gpx3 spectra can be attributed to the fluorescence of Gpx3 itself and scattering caused by the protein structure. The background and fold-change in fluorescence is similar in magnitude to that observed for other biological systems and probes described in Nature Communication^{2,3}.

Reviewer 2:

1. “To test the specificity of the probes, the authors used small potentially reactive biomolecules (fig 2 i,j) showing almost no reactivity except for CSA. GSH is still low but slightly elevated – given the concentration of GSH in cells, could this be a potential issue, considering that PTMs typically have a low stoichiometry?”

The reviewer astutely points out the minor increase CysOx fluorescence signal in the presence of millimolar GSH. The minor increase in signal is not due to cross-reactivity of probe with reduced GSH, rather it stems from the reaction of CysOx probe with GSH-SOH, which is formed by reaction of H₂O₂ with the GSH thiol (note that H₂O₂ forms in buffer by the reaction of O₂ with trace metal ions). To unambiguously demonstrate this point, we incubated 50 μM CysOx probe with 10 mM GSH with or without catalase, a H₂O₂-metabolizing enzyme. No significant increase in CysOx1 or CysOx2 fluorescence is observed. Furthermore, the reaction between GSH and H₂O₂ is slow (1 M⁻¹s⁻¹). Thus, fluorescence signal resulting from GSOH and CysOx probe is also expected to be quite low. The new data, measured at 5 mM GSH, are fully consistent with this fact. This new data is included as Supplementary Fig. 6a-c of the revised manuscript

2. “Also, the text reads glutathione sulfinic acid had been used, while the figure reads methanesulfinic acid. What was the control?”

The control is glutathione sulfinic acid. Figure 2 has been corrected.

3. “...they demonstrate that reduced Gpx3 does not react with their probes, but what if this C36 was a sulfinic acid version...will the probe still be as selective as shown for the small biomolecules?”

As the reviewer points out, cysteine sulfinic acid does not react with CysOx1 or 2 (Fig. 2i,j). The sulfur atom in sulfinic acid is not of sufficiently electrophilic in nature to react with carbon nucleophiles such as that present in CysOx probes⁴. Additional experiments with Gpx3 C36S (R4 below) further illustrate selectivity.

4. “...is Gpx3 quantitatively labeled? The authors should mention the efficiency of the labeling directly in the figure.”

As can be seen from the mass spectra in Fig. 3 (left panels), Gpx3 is quantitatively labeled by CysOx1 and CysOx2 probes. Our intact MS experiments with Gpx3 show all mass signals within a large range, it is not a SMIM analysis. The reactions in Fig. 2f,g were similarly analyzed by intact MS and determined that approximately 75% of Gpx3 is labeled by CysOx probes in these experimental conditions. We have clarified this in the figure legend, as requested.

5. “The probes show minimal cytotoxicity, which is great. I wonder what side effects the treatment with such probes will have though. What would a differential proteomic study with and without probe treatment (as performed in the kinase inhibitor screen) reveal, and what would this implicate?”

When applied at mM concentration, we have shown that the related nucleophile, dimedone inhibits signaling in a yeast oxidative stress pathway⁵. Our fluorogenic probes are used at μM concentration and are, thus, unlikely to have a similar effect. Moreover, our chemoproteomic studies show that CysOx probes do not

perturb the cysteinome. These new data are included as Supplementary Table 2 and Supplementary Fig. 13 in the revised manuscript. The above additional studies suggested by the reviewer, while intriguing, are beyond the scope of the present manuscript.

- 6. “Please give some more details about the kinase inhibitor library, e.g., what is known about the specificity of these inhibitors?”**

The Kinase Screening Library was purchased from Cayman Chemical Company. The Kinase Screening Library consists of two plates and contains approximately 160 carefully chosen selective and non-selective kinase inhibitors in a 96-well Matrix™ tube rack format as 10 mM stock solutions in DMSO. This curated library includes inhibitors of a wide range of lipid, receptor and non-receptor tyrosine, serine/threonine, and dual specificity kinases including those belonging to the ROCK, activin-like kinase (ALK), GSK3, PKC, PDGFR, VEGFR, Src, MAPK, CDK, and PI3K families, among many others. It offers expansive coverage, targeting more than 70 distinct kinases and kinase families, as well as numerous additional kinase isoforms and individual kinases within target families. Detailed information on inhibitor selectivity can be found on the manufacturer website HERE. This information has been included in the revised Methods section.

- 7. Huang et al. (PMID 27281782) have previously described the impact of kinase stimulation on Cys modifications, including the modulation of Cys in the active sites of phosphatases. Same holds true for Cys-nitrosylation (PMID 31097712). This should be added to the discussion, as these results are in good agreement with the finding that kinase inhibition affects Cys redox modifications.**

These findings are discussed in the revised text on p. 16.

- 8. Page 10, line 182: “suggesting a relationship between this idiosyncratic kinase and protein S-sulfenylation.” Page 12, line 203 “These findings suggest that redox changes induced by GSK3 inhibition might be more specific as opposed to simple diffusion of H₂O₂.” It is known that the inhibition of GSK, e.g., by SB-216763 is involved in the regulation of oxidative stress. This should be considered and mentioned here. Especially the second statement does not surprise in this context.**

These findings are discussed in the revised text on p. 15.

- 9. “Figure 5: What exactly does it mean that “the purity of the inhibitors was verified by LC-MS”? That the purity was as claimed by the manufacturer?”**

Yes, we analyzed inhibitors by LC-MS to verify that purity was as claimed by the manufacturer. This is now explicitly mentioned in the revised text.

- 10. “The chemo-proteomics results part would benefit from 2-3 more lines on what has actually been done. Also, the data should be uploaded to a repository, as is standard in the field. Considering a 1-hour treatment, did the authors consider the impact of protein expression level changes induced by the treatment? For phosphorylation it has been shown by the Gygi lab that a substantial share of the supposedly regulated phosphorylation events were indeed rather changes on the level of protein expression. Although I do not consider this as a major issue here, it would be good to briefly discuss this possibility.”**

Additional text to detail the chemoproteomics study and qualifying statement regarding protein expression have been added on p. 12-13 and again on p. 15 of the revised text.

The chemoproteomic dataset has been deposited to the ProteomeXchange Consortium via the PRIDE partner repository with the dataset identifier PXD029176. Data access connectivity in the ProteomeXchange: <http://proteomecentral.proteomexchange.org/cgi/GetDataset?ID=PX029176>. Data access in iProX link: <https://www.iprox.cn/page/project.html?id=IPX0003613000>.

- 11. “Figure 6: When looking at the suppl. data, it appears that some sites have only been quantified in a single replicate --- it is unclear whether these are included in the figure and in the results in general. If so, they should be removed and the discussion should focus on findings that have been replicated. In general, the suppl. Table is not easy to read and should be changed to a more user-friendly format (e.g., individual replicate ratios should be in individual cells, not separated by “|”). Did the authors use any site localization assessment to ensure that sites in peptides with multiple Cys residues are correctly assigned?”**

The reviewer is correct – some sites have been quantified in a single replicate. This is due to the stochastic nature of DDA-based shotgun proteomics. Such so-called “single-hit-wonders” have been removed when generating the corresponding figure and in our bioinformatics analyses. Furthermore, as suggested by Reviewer 3 (see below), we now use volcano plots (more reliable indicators) to show changes in the HeLa-sulfenome with GSK3 inhibition. We have reformatted Supplemental Table 4 as requested. Also, since the informatic pipeline that we used cannot provide automatic site localization assessment, probe-modified sites in peptides with multiple cysteines are manually evaluated and curated as described previously⁶. This process has been detailed in the revised Method section.

- 12. “The manuscript title should be changed to “cysteine sulfenic acid” rather than “cysteine oxidation” as this is actually misleading, unless the authors clearly explain how this same probe(s) can be used for other Cys oxidative PTMs?”**

We appreciate the reviewer’s point. At the same time, we feel strongly that this an issue of semantics. The probes do, in fact, monitor cysteine oxidation. We prefer to keep the title as is because readers outside the field may not be familiar with cysteine sulfenic acid. We are hoping to reach nonexperts through publication in Nature Communications. With a cursory read of the abstract, readers interested in cysteine oxidation will quickly understand that we are monitoring cysteine oxidation as cysteine sulfenic acid.

- 13. Suppl. methods: Why was sodium ascorbate used in the BTD assay, please explain. Shouldn’t this reduce Cys nitrosylation?**

Sodium ascorbate is used to recycle the Cu catalyst in the click chemistry reaction. Sodium ascorbate can reduce Cys-SNO to Cys-SH; neither of these species (RSNO or RSH) react with CysOx probes (Figs. 2-3, Supplementary Fig. 6).

Reviewer 3:

- 1. “The consistent identification of GSK3-beta inhibitors from the small molecule screen is intriguing. However, as no kinase inhibitor is exquisitely selective, one worries about potential off-target effects. Can the authors provide some reassurance that the effect of these inhibitors to promote sulfenic acid modification is on-target? For example, the three compounds vary in their efficacy in promoting Cys modification. Does this effect correlate with their potencies as GSK3-beta inhibitors? Can the authors verify target engagement by correlating a marker for GSK3-beta inhibition (e.g. loss of beta-catenin phosphorylation) with efficacy in inducing probe reactivity in their cell culture system?”**

These are excellent questions that led us to perform additional analyses detailed below. Please also see our response above to Reviewer 1, comment 2.

GSK3 inhibitor potency: In our experiments, three GSK3 inhibitors (BIO, BIM-IX, and SB-216763) were identified as being associated with increased S-sulfenylation in cells (3.3 to 9.7-fold). The IC₅₀ of these individual inhibitors against GSK3 α/β are similar, <10 nM⁷. The IC₅₀ for each structural class of inhibitors is as follows: indirubines, (BIO; GSK3 α/β IC₅₀<600 nM), bisindolyl maleimides (BIM-IX; GSK3 α/β IC₅₀<140 nM), and indolyl-maleimides (SB-216763; GSK3 α/β IC₅₀<35 nM)⁷. Overall, “compound class potency” correlates well with the observed increase in protein S-sulfenylation with BIO < BIM-IX < SB-216763. In our studies, we found that SB-216763 promoted the greatest increase in protein S-sulfenylation (Figs. 4-5).

This compound is a potent and selective inhibitor of GSK3 α/β ⁸, while BIO also inhibits cyclin-dependent kinases⁹ and BIM-IX also inhibits protein kinase C isoforms¹⁰. Thus, while IC₅₀s for the three inhibitors are similar, they are partitioned differentially among cellular targets. Furthermore, inhibitor EC₅₀s for β -catenin phosphorylation correlates well with GSK3 inhibitor selectivity and increased S-sulfenylation. We summarize this detailed information and emphasize the relationship between these inhibitor classes in the Discussion on p. 15 and Supplemental Fig. 15 of the revised manuscript.

2. ***“In the proteomics experiments, cutoffs used to score modification induced by inhibitor treatment are too loose – for the less potent inhibitors a “hit” is anything with a modification ratio >1.0, which of course is going to include a lot of sites that are not significantly upregulated. It would be preferable if they used both an effect size cutoff AND a significance cutoff (-log p > 1.3 is standard). Doing so will almost certainly reduce the number of consistent hits, which should make the GO analysis more robust. It would also be of interest to know how the GO analysis compares with prior studies that have used general oxidizing agents to induce sulfenic acid modification – in other words, are GSK3-beta inhibitors having a selective effect as suggested by the authors, or do they simply induce a low level of oxidative stress (which would still be interesting).”***

We thank the reviewer for raising this issue and for the suggestion of how to better illustrate and analyze our data. We now present the data in Fig. 5c as volcano plots, which allows us to detect changes in a more reliable fashion. In addition, as suggested by the reviewer, we compared the H₂O₂-dependent sulfenylome dataset from our earlier work¹¹. These studies are compared on p. 15 of the revised text, as requested.

3. ***“On p. 10, the references to Fig 5 are mislabeled as Fig 4.”***

This has been corrected.

Reviewer 4:

1. ***“Figures 3de and 3hi: It is important to show that H₂O₂-induced Gpx3-SOH formation is inhibited in Gpx3-C36A mutant, but not Gpx3-WT, detected by CysOx1 and CysOx2 probes using in gel fluorescence analysis.”***

We have performed the requested experiments as well as intact MS analysis. The resulting data show that H₂O₂-induced Gpx3-SOH formation, as detected by CysOx probes, is inhibited for Gpx3-Cys36Ser mutant, as expected. These new data are presented in Supplementary Fig. 7 of the revised manuscript.

2. ***“Figure 4C: Data should be quantified by showing averaged graphs with dot-plots and statistical analysis.”***

We have quantified the data in Fig. 4c, as requested. The resulting plot reports on the average fluorescence per lane obtained from three independent experiments and is included as Supplementary Fig. 9e in the revised manuscript.

3. ***“Before investigating effects of kinase inhibitors on cysteinyl oxidation in cells, authors should confirm their previous observation that H₂O₂ induces oxidation of EGFR at Cys797 (Cell Chem Biol 2016. 23:837-) in cells transfected with EGFR-WT vs. EGFR-Cys797A using CysOx1 and CysOx2 probes.”***

This is an excellent suggestion that led us to perform additional experiments. Cells were transfected with plasmids encoding for epitope-tagged EGFR-WT or EGFR-C797S. H₂O₂-treatment of transfected cells treated with CysOx probes led to the observation of in gel fluorescence for WT, but not C797S EGFR, as expected. These new data are included as Supplementary Fig. 8a,b in the revised manuscript.

4. ***“In Line 196-199, it is mentioned that “the sub-sulfenylome perturbed by GSK3 inhibition was enriched in many biological processes involved in antioxidant responses including regulatory cysteines of thioredoxin, thioredoxin reductase, glutaredoxin and glutaredoxin reductase (Fig. 5de). These data are presented in Figure 6. It is important to provide the number of experiments and statistical analysis.”***

The number of experiments and statistical analysis is now indicated in the legend to Fig. 6.

5. ***“Authors should provide the functional significance of s-sulfenylation of identified antioxidant enzymes in GSK-dependent biological responses such as cell survival or cell proliferation or cell migration in H₂O₂-treated HeLa cells by transfecting mutant proteins in which reactive Cys residue is mutated to Ala or Ser.”***

The above suggested studies, while important to dissect the precise molecular relationship between GSK3 inhibition and redox signaling, are beyond the scope of the present manuscript, which is focused on the development, selectivity, and cellular application of the very first fluorogenic probes for detecting cysteine oxidation.

6. ***“Since GSK is kinase, direct substrates should be phosphorylated. Thus, it is important to clarify the relationship between phosphorylation and s-sulfenylation of antioxidant enzymes, too.”***

Please see our response to comment #5 above.

7. ***“Supplementary Figure 7: It is important to provide the lists of s-sulfenylated proteins localized in ER and mitochondria, respectively.”***

This list has been included.

8. ***“Supplementary Figure 12: It is important to provide the number of experiments and statistical analysis.”***

The requested numbers are now indicated.

9. ***“There are lots of mislabeling of Figures, which should be corrected.”***

This has been corrected.

References

1. Fu, L., Liu, K., Ferreira, R. B., Carroll, K. S. & Yang, J. Proteome-wide analysis of cysteine S-sulfenylation using a benzothiazine-based probe. *Curr. Protoc. Protein Sci.* 95, e76 (2019).
2. Barth, N. D. *et al.* A fluorogenic cyclic peptide for imaging and quantification and drug-induced apoptosis. *Nat. Comm.* **11**, 4027 (2020).
3. Klima, J. C. *et al.* Incorporation of sensing modalities into de novo designed fluorescence-activating proteins. *Nat. Comm.* **12**, 856 (2021).

4. Akter, S. *et al.* Chemical proteomics reveals new targets of cysteine sulfinic acid reductase. *Nat. Chem. Biol.* **14**, 995–1004 (2018).
5. Paulsen, C. E. & Carroll, K. S. Chemical dissection of an essential redox switch in yeast. *Chem. Biol.* **16**, 217-225 (2009).
6. Yang J, Gupta V, Carroll KS, Liebler DC*. Site-specific mapping and quantification of protein S-sulphenylation in cells. *Nat. Commun.* **5**, 4776(2014)
7. Kramer, T., Schmidt, B. & Lon Monte, F. Small-molecule inhibitors of GSK-3: Structural insights and their application to Alzheimer's disease models. *Int. J. Alz. Dis.* 2012, 381029 (2012).
8. Coghlan. M. P. *et al.* Selective small molecule inhibitors of glycogen synthase kinase-3 modulate glycogen metabolism and gene transcription. *Chem. Biol.* **7**, 793-803 (2000).
9. Hers, I., Tavaré, J. M. & Denton, R. M. The protein kinase C inhibitors bisindolylmaleimide I (GF 109203x) and IX (Ro 31-8220) are potent inhibitors of glycogen synthase kinase-3 activity. *FEBS Lett.* **460**, 433-436 (1999).
10. Meijer *et al.* GSK-3-selective inhibitors derived from tyrian purple indirubins. *Chem. Biol.* **10**, 1255-1266 (2003).
11. Akter, S. *et al.* Chemical proteomics reveals new targets of cysteine sulfinic acid reductase. *Nat. Chem. Biol.* **14**, 995–1004 (2018).

REVIEWERS' COMMENTS

Reviewer #1 (Remarks to the Author):

Although authors added incremental explanation for this high background fluorogenic probe, I think these results do not verify any usefulness in biological applications. I think this paper does not meet high quality of this Journal.

Reviewer #2 (Remarks to the Author):

The authors have addressed comments adequately. I recommend acceptance of the revised manuscript.

Reviewer #3 (Remarks to the Author):

The authors have addressed all of my concerns with the original manuscript.

Reviewer #4 (Remarks to the Author):

Authors responded to most of this reviewer's concern. However, there is still major weakness that they failed to show the functional significance of GSK3 inhibitor-induced key s-sulfenylated proteins (antioxidant enzymes) in this study.

Reviewer comments and revisions:

Reviewer 1 (Remarks to the Author): Although authors added incremental explanation for this high background fluorogenic probe, I think these results do not verify any usefulness in biological applications. I think this paper does not meet high quality of this Journal.

Response: The utility of these probes in fluorogenic detection of sulfenic acid has been demonstrated in a variety of assays both *in vitro* and in cells. We have clearly discussed the limitations of these reagents in the final paragraph of the Discussion section as follows:

“As with all technologies, it is worth noting current limitations. At present, the CysOx fluorophore presents low brightness and quantum yields when compared to other dyes, such as fluorescein and BODIPY. This is reflected in the low quantum yields (0.9-2.4%) observed for the reaction product of sulfenic acid and CysOx probes. While this issue did not hinder application of CysOx probes in the present work, attempts to further miniaturize the cell-based assay to a 384-well plate format were unsuccessful. Future iterations of these tools will involve improving probe brightness and dynamic range. We also note that the ability of CysOx probes to monitor subtle endogenous signaling processes was not examined in this study but will be an important area of future research.”

Reviewer 2 (Remarks to the Author): The authors have addressed comments adequately. I recommend acceptance of the revised manuscript.

Reviewer 3 (Remarks to the Author): The authors have addressed all of my concerns with the original manuscript.

Reviewer #4 (Remarks to the Author): Authors responded to most of this reviewer's concern. However, there is still major weakness that they failed to show the functional significance of GSK3 inhibitor-induced key S-sulfonylated proteins (antioxidant enzymes) in this study.

Response: Elucidation of the functional significance of GSK3 inhibitor-induced S-sulfonylated proteins is beyond the scope of the current study and is an intriguing area of future research.